# HOW LLMS LEARN TO REASON: A COMPLEX NETWORK PERSPECTIVE

## ABSTRACT

Training large language models with Reinforcement Learning with Verifiable Rewards (RLVR) exhibits a set of distinctive and puzzling behaviors that remain poorly understood, including a two-stage learning curve, a V-shaped response-length trajectory, and a pronounced vulnerability to catastrophic forgetting. In this work, we propose that these behaviors are emergent collective phenomena governed not by neural implementation details, but by the topological evolution of the latent reasoning graph in semantic space. By demonstrating a dynamical isomorphism between a 1.5B-parameter LLM and a minimal Concept Network Model (CoNet), we trace the causal source to the self-organization of a sparse concept web pinned to an average degree of two. This geometric perspective provides a unified physical explanation for the observed anomalies: the V-shaped trajectory tracks the evolution from parallel local skill optimization to global network integration; catastrophic forgetting stems from the topological disconnection of critical "trunk" edges; and policy collapse arises from the accumulation of sequential transitions at the web's leaf nodes, where broad exploration abruptly freezes into rigid, high-reward trajectories. Identifying a "maximally frustrated state" at the transition between learning stages, we propose *Annealed-RLVR*, a principled algorithm that injects a targeted SFT "heating" step to resolve this topological bottleneck. Experiments confirm that this theory-driven intervention outperforms standard RLVR on both in-distribution and out-of-distribution benchmarks (including Minerva and AIME). By recasting RLVR from black-box optimization into a predictable process of structural self-organization, our work provides a new physical intuition for engineering the emergent reasoning capabilities of future AI systems.

## 1 INTRODUCTION

Training large language models with reinforcement learning with verifiable rewards (RLVR) (Liu et al., 2025; Guo et al., 2025; Team et al., 2025; Yang et al., 2025) has revealed four striking and still poorly understood phenomena. On math and coding benchmarks, RLVR runs typically show (i) rapid initial gains followed by a long performance plateau, even as training continues (Luo et al., 2025a; He et al., 2025; Jin et al., 2025b); (ii) a V-shaped trajectory of the length of correct solutions, where answers first become markedly shorter before later growing longer again (Luo et al., 2025a; He et al., 2025; Fatemi et al., 2025; Wang et al., 2025); (iii) pronounced vulnerability to catastrophic forgetting when RLVR-trained policies are interleaved or followed by supervised fine-tuning (SFT), with earlier reasoning skills abruptly degraded (Li & Hoiem, 2017; Luo et al., 2025b; Ding & Wang, 2025); and (iv) a collapse of solution diversity, with the policy converging onto a narrow and brittle solution manifold (He et al., 2025; Cui et al., 2025; Hao et al., 2025; Jiang et al., 2025). Existing work offers only partial, largely phenomenological explanations for each phenomenon in isolation: plateaus and the collapse of solution diversity are linked to entropy exhaustion and over-exploitation of a fixed reward signal (Cui et al., 2025; Hao et al., 2025; Jiang et al., 2025); the initial shortening of solutions is attributed to reward-driven pruning of redundant reasoning, whereas the

subsequent lengthening has been ascribed either to a shift toward solving harder instances that demand longer chains of thought, or to the emergence of multi-pass, self-reflective reasoning ("aha"-like behavior) under RLVR(Fatemi et al., 2025; Wang et al., 2025); and catastrophic forgetting is analyzed as an objective-mismatch problem between RLVR and subsequent SFT or other post-training stages (Luo et al., 2025b; Ding & Wang, 2025). However, these accounts remain fragmented: we still lack a unifying framework that ties all four behaviors to a common underlying structure or dynamical mechanism of RLVR-trained policies, and explains why they so often arise together in practice.

A growing body of work tries to explain these phenomena by modeling LLMs reasoning process as an implicit reasoning graph—a network of semantic states and logical transitions (Wang et al., 2024; Dutta et al., 2024; Xiang et al., 2025; Cabannes et al., 2024; Minegishi et al., 2025). However, translating this idea into a fully microscopic analysis has proven extremely difficult: constructing the reasoning graph from a model's high-dimensional latent space is a formidable open problem, preventing a direct investigation into the structural origins of the observed dynamics (See Appendix C for details).

Here, we argue that the very universality of the above RLVR dynamics provides a path forward. In complex systems, collective emergent behaviors are largely independent of microscopic details and governed by the system's large-scale organization (Anderson, 1972; Watts & Strogatz, 1998; Barabási & Albert, 1999). This insight motivates an approach rooted in the core logic of the Renormalization Group (RG) (Wilson & Kogut, 1974): instead of directly studying the system at the full microscopic level, one can model and approach the system at a coarse-grained level to explain its macroscopic behavior. Applying this strategy, we shift our focus from the full reasoning graph at the token level to its coarse-grained backbone at the semantic level, which emerges by abstracting away fine-grained skills encoded in local network patterns like motifs or community structures (Milo et al., 2002; Fortunato, 2010; Zhang & Moore, 2014). This leads us to our central hypothesis regarding the emergent structure that forms under RLVR training for general-purpose System-2 reasoning:

> **Hypothesis 1.** *The **concept web**, defined as the coarse-grained reasoning graph, is a sparse network whose effective average degree is pinned to $\langle k \rangle \approx 2$.*

The implications of this hypothesis are non-trivial: an average degree of 2 implies a predominantly tree-like structure, which is efficient for generalization but inherently fragile. This topology can still support rare, large-scale cycles critical for complex operations like self-correction (as detailed in Appendix C), without significantly altering the global average degree.

To investigate this hypothesis, we adopt a comparative approach connecting macroscopic phenomenology to graph topology. We first confirm that the DeepSeek-R1-Distill-Qwen-1.5B model (Luo et al., 2025a) exhibits the aforementioned RLVR signatures: a two-stage learning curve, a V-shaped response trajectory, catastrophic forgetting, and policy collapse. We then show that these complex behaviors are qualitatively reproduced by the Concept Network Model (CoNet) (Cai et al., 2025), a minimal framework abstracting away neural implementation details. This striking resemblance suggests that these phenomena are emergent collective behaviors rather than artifacts of specific neural architectures, allowing them to be studied via a coarse-grained proxy. Employing CoNet, we identify the emergence of a sparse concept web that orchestrates these dynamics. This phenomenological alignment supports the hypothesis that similar topological self-organization underlies LLM reasoning, enabling us to derive the following mechanisms from the principles of sparse graph growth:

First, the V-shaped response length implies a structural evolution from local "skill islands" to a global concept web. The initial decrease results from the parallel optimization of independent skills, where the policy rapidly discards inefficient exploratory detours to converge on short, optimal paths for simple tasks (Fig. 2). The subsequent rise, however, marks the onset of global integration, where a tree-like web emerges (Fig. 3). On such a sparse substrate, the average geodesic distance is inherently correlated with network size. Thus, as

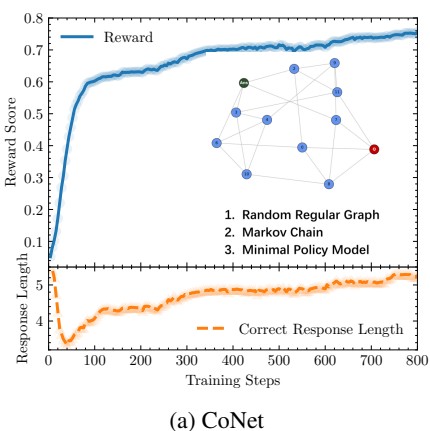
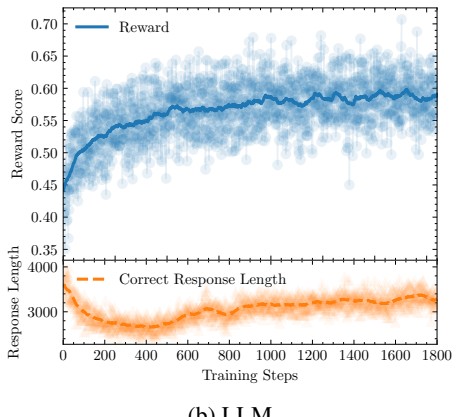

(a) CoNet                                      (b) LLM

Figure 1: **CoNet Reproduces Core LLM Training Dynamics.** The minimal CoNet model (a) reproduces the two core empirical signatures of RLVR training observed in the DeepScaleR-1.5B LLM (b). These signatures are: (i) a two-stage reward dynamic consisting of a fast-learning stage followed by a slow-learning phase (top panels), and (ii) a non-monotonic, V-shaped evolution of the correct response length (bottom panels). This striking correspondence validates CoNet as a valuable minimal model for our theoretical analysis. The inset in (a) schematically depicts the CoNet abstraction. Lines in (b) are smoothed with a 60-step moving average.

the web expands to bridge distant islands during the slow-learning phase, the policy is geometrically forced to traverse longer paths. The rising V-curve is therefore not a change in reasoning style, but the result of the concept web's continuous growth.

Second, the sparse topology explains the model's specific vulnerability to catastrophic forgetting (Fig. 4). Unlike dense networks, a sparse web relies on critical "trunk" edges to support vast downstream branches. We show that SFT-induced forgetting acts as a topological disconnection: gradient updates overwrite weights at these high-traffic bottlenecks, severing bridges to entire sub-trees. This offers a geometric rationale for the "easy-to-break, easy-to-fix" nature of RLVR models; since the disconnected downstream knowledge remains intact, a brief RLVR resumption leads to rapid recovery by merely "re-soldering" the broken trunks rather than relearning the web.

Third, we trace policy collapse to the dynamics of learning at the web's frontier (the leaf nodes). Extending the critical learning theory of Cai et al. (2025), our experiments suggest that during the slow-learning phase, a significant portion of tasks sequentially undergo continuous-phase-transition-like learning dynamics (Fig. 5). For these tasks, the policy shifts from broad exploration to locking onto specific solution paths. While this collapse is necessary for mastering individual skills, standard RLVR accumulates these local entropy reductions across the leaf nodes. This aggregation ultimately results in a global loss of diversity, manifesting as the policy collapse observed in reasoning benchmarks (Fig. 6).

Finally, identifying a "maximally frustrated state" at the onset of slow learning—where competition between disconnected islands peaks—motivates our Annealed-RLVR algorithm. By introducing a targeted SFT "heating" step at this topological bottleneck, we temporarily relax competitive tension, helping the system navigate past local optima before cooling into a refined structure. Our experiments confirm that this theory-driven approach resolves the bottleneck and achieves superior reasoning performance on in-distribution and out-of-distribution benchmarks (including Minerva and AIME), surpassing standard RLVR baselines by engineering the topology of reasoning.

## 2 THE V-SHAPED SIGNATURE AS EVIDENCE FOR A SPARSE CONCEPT WEB

The V-shaped evolution of the response length is a well-demonstrated, quantifiable hallmark of deep structural reorganization during RLVR training (He et al., 2025; Luo et al., 2025a). To anchor our investigation, we first reproduce this signature by applying RLVR to the DeepSeek-R1-Distill-Qwen-1.5B model following the DeepScaleR protocol (Luo et al., 2025a). As shown in Fig. 1, our experiment recovers the characteristic V-shaped curve (Fig. 1).

To dissect the mechanism underlying this signature—a task intractable within the LLM itself—we deploy an extended, multi-task variant of the CoNet as a "computational microscope" (See Appendix D for details). CoNet instantiates the implicit reasoning graph discussed in the Introduction as a concrete, tractable model: it maps the LLM "semantic states" to abstract nodes in a fixed random graph and LLM "logical transitions" to learnable, probabilistic edges. This minimal model reduces the learning process to a tractable graph-traversal problem and, remarkably, reproduces the V-shaped curve observed in our LLM experiment (Fig. 1), validating it as a faithful proxy for uncovering the underlying mechanism. Armed with this confirmation, we now demonstrate that the V-shaped evolution of the response length arises from a two-stage process. The decreases of the response length in the initial fast-learning stage is dominated by local optimization. Given the random initialization of transition probabilities, shorter paths inherently have a higher probability of being sampled. Consequently, the model rapidly discovers short, efficient solutions to individual problems, giving rise to a proliferation of disconnected "skill islands" (as shown in Fig. 2(a)). This burst is quantified by the number of distinct solution clusters, which peaks right before the onset of slow learning (Fig. 2(b)). Stage one thus constitutes a parallel search for local efficiency that naturally shortens the average response length.

The subsequent slow-learning phase, where response length steadily increases, marks a fundamental shift to global integration. The primary task is no longer discovering new islands but weaving them into a single, expansive concept web (see Fig. 2(c)), defined to be the largest connected component in which every edge has a transition probability above 0.95. As the concept web relentlessly grows, a critical structural property

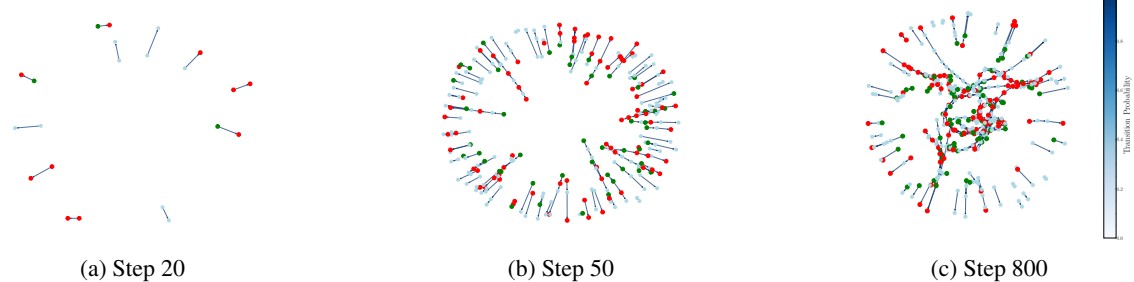

(a) Step 20       (b) Step 50       (c) Step 800

Figure 2: **Visualizing the Structural Evolution from "Skill Islands" to the Concept Web.** These network snapshots provide a direct, qualitative narrative of the concept web's formation, corresponding to the different phases of training. In each subfigure, green, red, and blue dots indicate the question, answer, and intermediate nodes of the CoNet, respectively. The color of each directed edge represents the transition probability from the head to the tail and is drawn according to the colorbar in the rightmost column. Here, only edges with transition probabilities greater than 0.95 are shown. **(a)** In the early fast-learning phase (Step 20), the model discovers a few short, disjoint reasoning paths, representing the first nascent "skill islands". **(b)** At the onset of slow learning (Step 50), the system has proliferated into a maximal collection of disconnected islands. **(c)** Deep in the slow-learning phase (Step 800), these previously separate islands have coalesced into a single, giant connected component, forming the unified and expansive concept web.

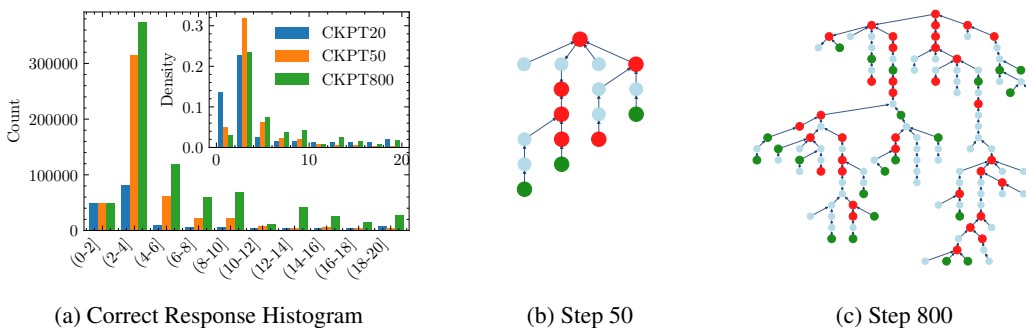

(a) Correct Response Histogram         (b) Step 50         (c) Step 800

Figure 3: **A Sparse-Web Structure Necessitates Longer Reasoning Chains.** This figure links the emergent sparse topology of the concept web to the observed increase in response length. (b, c) The color and marker scheme is identical to that used in Fig. 2, and only edges with transition probability greater than 0.95 are retained. The largest connected component (a local region of which is shown here) remains sparse with average degree around 2 even as it grows from step 50 to 800. (a) Consequently, the distribution of solution path lengths, shown in both raw counts (histogram) and probability density (inset), shifts decisively to the right during this period, confirming that navigating the sparse backbone requires longer reasoning chains.

emerges: the web remains persistently sparse, with its average degree stabilizing near two (as shown in Fig. 3(c)). This provides direct, quantitative validation for our central Hypothesis (Hypothesis 1).

This emergent sparsity is the topological key to the V-shape's rising slope. A network with such a low average degree is dominated by chain and tree-like structures, inherently lacking the shortcuts and redundant pathways found in denser graphs. Consequently, connecting two previously distant concepts requires traversing long, intermediate paths. The histograms in Fig. 3(a) provide clear evidence of this effect: As the sparse web expands, the distribution of solution lengths undergoes a decisive and sustained rightward shift, confirming that navigation on this sparse backbone necessitates longer reasoning chains. Thus, the V-shaped curve furnishes a robust macroscopic signature of an underlying topological shift: from parallel, local optimization on disconnected components to serial, global navigation across an expanding, sparse backbone.

Yet can this simple, largely tree-like model capture the full complexity of LLM reasoning? After all, an LLM's autoregressive memory enables sophisticated behaviors such as self-correction or reflection (e.g., "Wait, let me rethink that...")—that would manifest as cycles in the reasoning graph. We argue they are fully compatible(Minegishi et al., 2025). These reasoning cycles are not the dense, ubiquitous shortcuts of a highly connected graph. Instead, they are likely the cognitive manifestation of sparse, local loops—occasional, high-cost revisions on a much larger, fundamentally chain-like scaffold. The existence of a limited number of such cycles does not significantly alter the global average degree of the web, which remains close to two. With the sparse-web model established, this link between a macroscopic dynamic and a microscopic topology forms the foundation for the analysis that follows.

## 3 THE DOUBLE-EDGED SWORD OF SPARSITY: FROM CATASTROPHIC FORGETTING TO SIMULATED ANNEALING SCHEME

From the graph theoretic point of view, the sparsity in the conceptual network implies low redundancy, fragility, and critical vulnerability. It also reflects a structural hypothesis for a well-known operational challenge: why models that have formed a concept web via RLVR are so susceptible to catastrophic forgetting

when subsequently fine-tuned with supervised learning (Li et al., 2024). Our theory posits that this is not a widespread erasure of knowledge, but a specific topological failure: the surgical severing of critical, bridge-like connections.

To test this mechanistic hypothesis, we subjected models in the deep slow-learning phase to a lightweight in-distribution SFT (see Appendix H and I). As expected, the intervention triggered a sharp drop in overall performance in both CoNet and the 1.5B LLM (Fig. 4(a), (b)). Having reproduced this known macroscopic phenomenon, we leveraged CoNets transparency to inspect the microscopic aftermath.

The result is a direct validation of our structural explanation. The microscopic view reveals that the SFT catastrophic forgetting was not a global degradation of the web. Instead, SFT's aggressive optimization of a few paths overwrote the policy at critical branching nodes, precisely breaking the single connection points for vast downstream subgraphs. Fig. 4(c), (d) provides a stark visualization of this: a unified web is instantly fragmented, offering a clear, structural cause for the catastrophic performance drop.

However, this is where the sword's second edge is revealed. The damage, while profound in its effect, is topologically localized. SFT does not erase the knowledge encoded in the now-disconnected subgraphs; it merely renders them unreachable. This localization explains the remarkable speed of recovery, which can

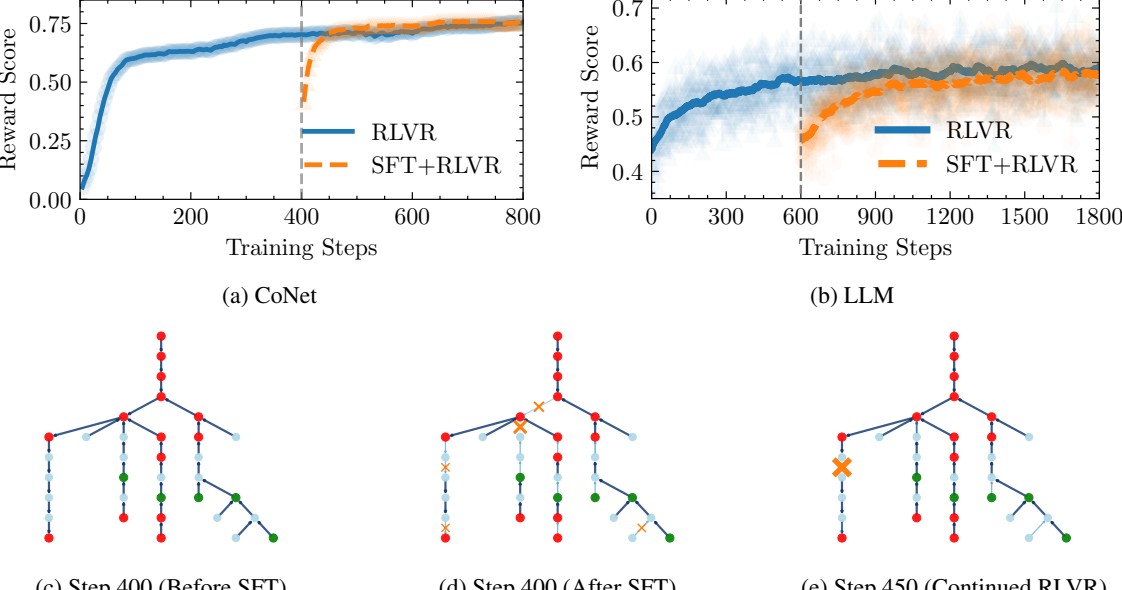

(a) CoNet          (b) LLM

(c) Step 400 (Before SFT)     (d) Step 400 (After SFT)     (e) Step 450 (Continued RLVR)

Figure 4: **The Fragility of a Sparse Web: Catastrophic Forgetting and Fast Recovery.** This figure demonstrates a key prediction of our sparse-web hypothesis: that the concept web is fragile, relying on critical bridge-like connections. (c-e) The color and shape conventions follow those of Fig. 2, and only a local view of the concept web is shown. Orange crosses on the edges indicate decreases in transition weights, with thicker crosses denoting more substantial decreases. The microscopic view in CoNet shows how a lightweight supervised fine-tuning (SFT) on a converged web (c) severs these critical bridges, causing the structure to fragment (d). Subsequent RLVR rapidly repairs these links (e). (a, b) This microscopic severing manifests as macroscopic catastrophic forgetting in both CoNet and the 1.5B LLM, where performance plummets upon initiating SFT. The subsequent fast recovery once RLVR resumes highlights the localized nature of the damage, underscoring the web's structural fragility.

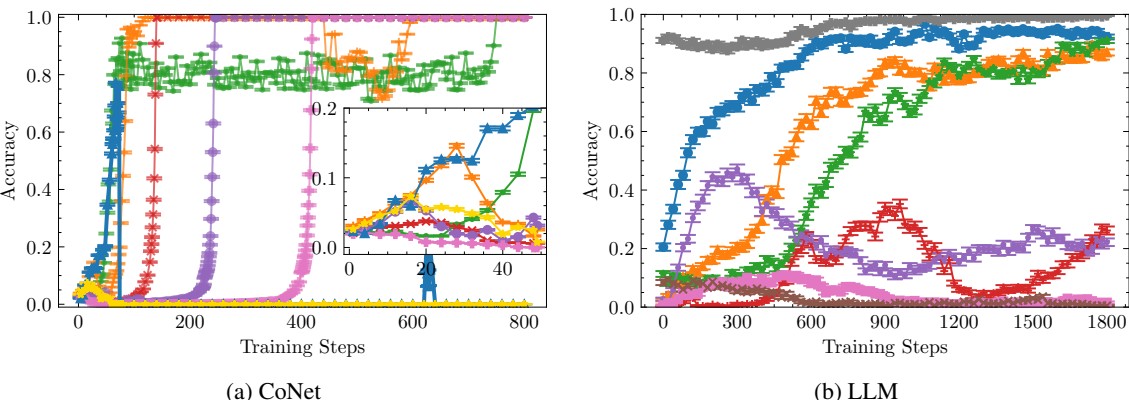

(a) CoNet                (b) LLM

Figure 5: **Microscopic Mechanisms of RLVR: Forgetting by Frustration, Learning by Phase Transition**
The learning trajectories for individual problems in CoNet (a) and the 1.5B LLM (b) reveal a fundamental
duality. (i) Frustration-Induced Forgetting: At the onset of slow learning [see inset in (a)], the intense
competition for connections on a sparse web manifests as volatile, non-monotonic accuracy curves, where
some skills are competitively suppressed. (ii) Phase-Transition-Like Learning: Subsequently, the web's
sparse frontier enables new skills to be acquired in punctuated, accelerated jumps (orange and green). The
smoother gradients in the LLM (vs. the clean steps in CoNet) can be attributed to finite-size effects, as
explained in the main text.

be observed in Fig. 4(a), (b). When RLVR training resumes, it triggers a quick rebound because the task is
simply to "re-solder" a few broken connections to restore access to the latent scaffold as shown in Fig. 4(e).

This two-step process of localized disruption followed by rapid repair is strongly analogous to simulated
annealing Kirkpatrick et al. (1983). The brief SFT phase acts as a high-temperature "heating" shock, break-
ing connections and allowing the system to escape a potentially suboptimal state. The subsequent RLVR
phase then acts as a controlled "cooling", guiding the system as it re-forms connections, ideally settling into
a more robust configuration (Lai et al., 2025). This insight—that a known vulnerability can be repurposed as
a deliberate annealing step—forms the foundation for the Annealed-RLVR algorithm, which we introduce
in Sec. 5.

## 4 Microscopic Mechanisms of RLVR: Forgetting by Frustration, Learning by Phase Transition

Although the overall reward curve in RLVR training looks smooth, our sparse-web theory shows that much
more is happening underneath. As shown in the individual trajectories in Fig. 5, the network's sparse struc-
ture creates two competing effects: (i) a forgetting process caused by frustration, which is strongest at the
start of the slow-learning phase, and (ii) an opposite, phase-transition-like learning process that gradually
builds and integrates knowledge during slow learning.

### 4.1 Frustration-Induced Forgetting: The Price of Integration

The first dynamic is driven by frustration-induced forgetting: a competitive process where the advancement
of some skills forces the deterministic regression of others as the model integrates its "skill islands". This
process manifests as volatile, non-monotonic learning curves in the learning dynamics (Fig. 5) and rep-

resents an internal "push-and-pull" fundamentally different from the external shock of SFT-induced catastrophic forgetting (Shenfeld et al., 2025; Jin et al., 2025a).

This process peaks in the maximally frustrated state at the onset of the slow-learning stage, a critical juncture where "skill islands" compete for limited connections under the $\langle k \rangle \approx 2$ constraint. Paradoxically, this state maximizes exploration: the model's ability to find diverse, correct out-of-distribution solutions peaks, far exceeding the base model or the same model after prolonged training (Fig. 6(b), blue). This subsequent degradation in performance is consistent with the phenomenon of policy collapse, which has been recently reported in several studies to occur after extended RLVR, leading to a loss of exploratory power (Chen et al., 2025; Yue et al., 2025; Cui et al., 2025).

### 4.2 PHASE-TRANSITION-LIKE LEARNING: THE ENGINE OF GROWTH

The opposing dynamic, phase-transition-like learning, is an important engine of new knowledge acquisition (Power et al., 2022; Lin et al., 2025). The evidence for this is clearly shown in CoNet's sharp, step-like accuracy jumps (Fig. 5(a)), and the steep accelerated learning gradients observed in the 1.5B LLM (Fig. 5(b), more details in Appendix G). This phenomenon is a multi-task extension of the "Learning at Criticality" (LaC) mechanism, first identified in a single-task context (Cai et al., 2025). The LaC framework models this process as a continuous phase transition (Stanley & Ahlers, 1972; Wilson, 1983) where the policy collapses from exploring diverse reasoning paths to exploiting a single optimal one. Crucially, analysis revealed that near this critical point, the system's exploratory power is maximized, and the reasoning path lengths become scale-invariant—a hallmark of continuous phase transitions. Beyond this transition, the policy collapses onto fixed optimal paths; the associated generalization drop is possibly linked to overfitting within the RLVR process (Dong et al., 2025; Gui & Cheng, 2025; Shypula et al., 2025; Zheng et al., 2025). This context explains the smoother gradient in the LLM. Compared to the step-like jumps in CoNet, the LLM's learning curve is a classic manifestation of finite-size effects (Fisher & Barber, 1972): it has a more constrained set of distinct reasoning paths for any given task, which rounds out the idealized sharp transition into a steep but continuous curve. In Appendix K, we reproduce the single-task training to isolate this phase transition behavior, providing a controlled setting for further investigation into its scaling behavior.

The puzzle is why this delicate phenomenon survives in a chaotic multi-task setting. Our sparse-web model explains this through topological isolation: a node on the expanding frontier of a $\langle k \rangle \approx 2$ network is statistically isolated, decoupling it from system-wide competition. Shielded from interference, a new skill can be efficiently integrated via a clean phase transition. This perspective also provides a microscopic explanation for aggregate policy collapse in the standard RLVR (Chen et al., 2025; Yue et al., 2025; Cui et al., 2025). We hypothesize this is not a monolithic failure but the statistical sum of asynchronous, microscopic collapses at the frontier, a hypothesis supported by the LaC finding that training just beyond criticality risks overfitting-induced collapse.

## 5 ANNEALED-RLVR: AN ALGORITHM BORN FROM THE SPARSE-WEB THEORY

Our sparse-web theory reveals a core narrative of RLVR training: the system drives itself into a maximally frustrated state that is both a perilous bottleneck and a moment of peak exploratory potential (Fig. 6(b), blue curve). This insight motivates Annealed-RLVR, a theory-driven intervention that leverages the high-entropy frustrated state to guide the model toward a more globally optimal configuration (Cheng et al., 2025; Chen et al., 2025).

The algorithm operationalizes this concept with a precisely timed intervention. This intervention is triggered by monitoring the macroscopic signatures of the maximally frustrated state, which, as shown in Fig. 1, correspond to the crossover regime characterized by the "knee" of the reward curve (when accuracy gains plateau) and the bottom of the V-shaped response length curve. At this juncture, a brief SFT "heating" phase

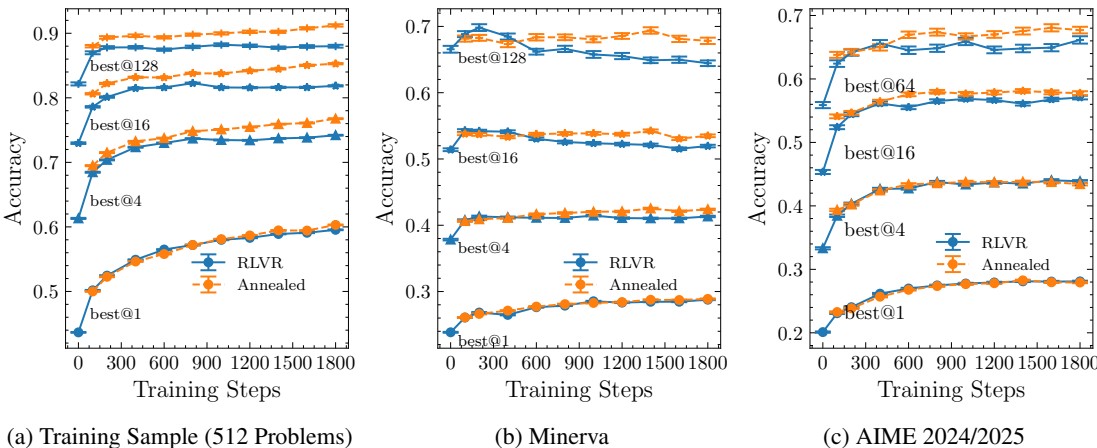

(a) Training Sample (512 Problems)  (b) Minerva  (c) AIME 2024/2025

Figure 6: **Annealed-RLVR Outperforms Standard RLVR.** This figure compares the best@k accuracy curves of our Annealed-RLVR ("Annealed") with the standard RLVR baseline ("RLVR"). The evaluation is performed on (a) an in-distribution set (Randomly Selected 512 Training Problems) and two out-of-distribution (OOD) sets: (b) the Minerva dataset and (c) the AIME 2024/2025 datasets. The results show that Annealed-RLVR consistently achieves superior performance across all in-distribution and OOD benchmarks, confirming that the intervention fosters improved generalization.

begins. The timing is critical: because the "skill islands" have not yet solidified into a single web, the system is uniquely robust to a targeted shock that would otherwise cause catastrophic forgetting—a theoretical claim confirmed by our experiments (Fig. 4). The SFT thus functions not as a disruptive overwrite but as a corrective jolt to break the policy's premature commitment to suboptimal routes. We implement this by surgically applying SFT only to problems with low accuracy (e.g., $< 0.1$) for which correct solutions exist. Immediately following this targeted heating, the "cooling" stage resumes standard RLVR, allowing the now more pliable policy to re-settle into a more robust and integrated final state.

We validated this approach on the DeepSeek-R1-Distill-Qwen-1.5B model, trained on the DeepScaleR-Preview-Dataset . Our method, Annealed-RLVR, was compared against the standard RLVR baseline where the policy is optimized using a variant of the Group Relative Policy Optimization (GRPO) algorithm (Shao et al., 2024; Liu et al., 2025; Yu et al., 2025). The results (Fig. 6) confirm our theory. Best@k accuracy curves demonstrate that Annealed-RLVR outperforms standard RLVR on both in-distribution evaluation set (512 randomly selected training problems) and on out-of-distribution (OOD) benchmarks, including the Minerva and AIME 2024/2025 dataset. The histograms (see Fig. 13) reveal the underlying mechanism: the SFT "heating" provides a crucial performance boost to the most difficult problems, lifting their success rates from near-zero and making them "visible" to subsequent RLVR exploration. The RLVR "cooling" phase then efficiently guides these newly accessible problems toward mastery. This results in a systematic reduction of unsolved problems and a corresponding population shift towards solved ones, confirming that our intervention guides the model past the inherent bottlenecks of standard RLVR to achieve superior reasoning. More details are given in Appendix H and I.

## 6 CONCLUSION AND OUTLOOK

We have introduced a sparse-concept-net theory to explain the puzzling behaviors of the RLVR training, including the two-stage learning curve, V-shaped response curves, and vulnerability to catastrophic forgetting.

Our theory proposes that these diverse phenomena are consequences of one underlying process: the formation of a sparse concept web, with the averaged degree pinned close to two. This physical model directly explains the V-shaped curve as a shift from local optimization to global integration and recasts catastrophic forgetting as the severing of the web's critical, bridge-like connections.

A natural consequence of this sparse structure is a fundamental RLVR mechanism for forgetting and learning within the microscopic training process. We find that the degree-2 constraint first forces the system into a maximally frustrated state at the onset of slow learning, a phase dominated by frustration-induced forgetting as skills are competitively suppressed. Subsequently, the same sparse structure enables a cleaner growth mechanism: phase-transition-like learning, where new knowledge is efficiently integrated at the web's expanding frontier , yet this very per-skill collapse onto a single high-reward path also accumulates over time into a macroscopic policy collapse and a global loss of solution diversity when training proceeds too far beyond criticality.

This framework is not just descriptive; it is prescriptive. It identifies the maximally frustrated state—the peak of competitive forgetting—as the ideal moment for SFT intervention. Our Annealed-RLVR algorithm is born from this insight. By applying a timed, corrective SFT shock, Annealed-RLVR mitigates the policy collapse and, as our experiments confirm, unlocks a superior final reasoning capability. Ultimately, this work provides a new lens for understanding RLVR: not as black-box optimization, but as a predictable process of structural self-organization, paving the way for more principled methods for engineering the advanced reasoning capabilities of future AI.

Looking forward, a central long-term goal is the empirical mapping of the complete, microscopic reasoning graph from large-scale foundation models. This remains a significant undertaking, requiring substantial computational resources and novel algorithms for semantic chunking and clustering. The payoff for this effort, however, would be transformative, producing a powerful new analytical object: a detailed map of an AI's internal world, composed of the reasoning graph and its emergent concept web. While the immediate application of this map would be to empirically test our sparse-web hypothesis, its long-term impact is far broader: charting this internal world is a critical step toward genuinely explainable LLMs and the robust safety of future artificial intelligence.

# 7 RELATED WORK

Our work intersects with three domains: training paradigms for LLM reasoning, the modeling of reasoning as a complex network, and the study of RLVR's failure modes. While building on the standard RLVR paradigm (Guo et al., 2025; Team et al., 2025; Yang et al., 2025), our goal is to provide a theoretical-physics-motivated theory for its perplexing macroscopic dynamics. To this end, we engage with the view of reasoning as a complex network; however, our approach differs from prior work that empirically extracts static reasoning graphs post-hoc (Xiang et al., 2025; Cabannes et al., 2024; Minegishi et al., 2025). Inspired by the core logic of the RG, we bypass the intractable graph-construction challenge and instead hypothesize a simple organizational principle for the emergent, coarse-grained concept web with $\langle k \rangle \approx 2$, enabling us to study the universal behaviros of RLVR. This sparse-web perspective provides a unified origin for seemingly distinct failure modes like catastrophic forgetting (Li & Hoiem, 2017; Ding & Wang, 2025) and policy collapse (Yue et al., 2025; Cui et al., 2025), framing the former as the severing of the network's fragile bridges, and the latter as a consequence of a phase-transition-like learning dynamics at its frontier (Power et al., 2022; Lin et al., 2025; Cai et al., 2025). An important line of research has made valuable progress on these failures using algorithmic solutions (Chen et al., 2025). Our work contributes a complementary, theory-driven perspective: we leverage the sparse-web model to design Annealed-RLVR, an intervention that targets the problem's underlying structural root. See Appendix B for extended related work.

## 8  REPRODUCIBILITY STATEMENT

To ensure full reproducibility, we provide a complete experimental blueprint in the Appendix: Sec. I details the LLM training pipelinemodels, hyperparameters, and SFT dataset construction, while Sec. J describes our evaluation protocol. Our code repository of CoNet can be available at: https://anonymous.4open.science/r/CoNet-83A4. The repository includes a README.md file that details the functionality of each key Python script. The training process is described in Appendix D and Appendix H.

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

## A  THE USE OF LARGE LANGUAGE MODELS (LLMS)

This work utilized Large Language Models (LLMs) to assist in the preparation of the manuscript, primarily for improving language clarity and readability. For the accompanying CoNet repository, we also employed LLMs to generate Python codes, such as the training algorithm, measuring network properties and visualizing the concept web and its largest connected component. All LLM-generated outputs, including text and code, were meticulously reviewed, edited, and verified by the authors to ensure their accuracy and suitability for this work.

## B  EXTENDED RELATED WORK

**System-2 Reasoning in LLMs.** The dual-process theory, distinguishing between fast, intuitive System-1 thinking and slow, deliberate System-2 reasoning (Kahneman, 2011), has become a valuable framework for analyzing LLM behavior. While standard LLMs often exhibit System-1-like responses, significant research has focused on eliciting deliberate System-2 reasoning. Prompting strategies such as Chain-of-Thought (CoT) (Wei et al., 2022) and its variants like Self-Consistency (Wang et al., 2022) and Tree of Thoughts (ToT) (Yao et al., 2023) encourage models to generate explicit reasoning steps. Other works aim to build this deliberative capacity into the model's architecture or training process through self-correction and reflection (Shinn et al., 2023; Madaan et al., 2023). Our work complements these efforts by shifting the focus from **eliciting** reasoning to understanding the **emergent mechanism** of how this capability develops internally during training.

**Complex Network Analysis of Neural Models.** Applying network science to understand neural networks is a long-standing idea, previously used to study the topology and robustness of smaller models (Simard et al., 2005; Estrada & Hatano, 2008). With the advent of Transformers, researchers have begun analyzing attention patterns as graphs (Vig, 2019) to better understand information flow (Abnar & Zuidema, 2020; Chefer et al., 2021). More recently, studies have explored the topological properties of neural networks, identifying optimal graph structures (You et al., 2020) and the formation of essential sparse subnetworks (Frankle & Carbin, 2019; Filan et al., 2021). Complementing these static analyses, recent work has modeled the training process itself as the evolution of complex networks (Danovski et al., 2024; Rocchetti, 2025). A recent study by (Wang et al., 2024; Minegishi et al., 2025; Matsutani et al., 2025) shares our goal of analyzing the network structures that underpin System-2 reasoning, though they do so by empirically extracting a graph from hidden-state representations. In contrast, our CoNet framework utilizes a fixed, abstract concept space, which allows us to specifically isolate and analyze the dynamic mechanism of how this reasoning structure emerges during training.

## C  THE REASONING GRAPH: FROM EMPIRICAL EXTRACTION TO A UNIFIED CONCEPT WEB

The idea of a reasoning graph has emerged in the literature as a powerful framework for understanding the chain-of-thought process in LLMs for a given task. This graph is not explicitly encoded in the Transformer's static parameters but is an emergent property of its inference dynamics. The standard model for its construction is based on a key observation: an LLM's generative process is guided by a few decision points of high Shannon entropy, or "forking tokens" (Fig. 7). This allows for a deconstruction of a chain-of-thought: stable, low-entropy text chunks form the graph's nodes (concepts), while the probabilistic choices at these high-entropy forks form the weighted, directed edges.

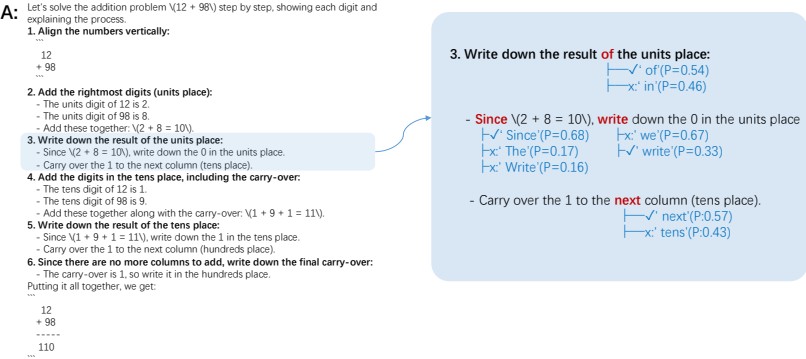

Figure 7: Illustration of the LLM reasoning process. A chain-of-thought is composed of stable, low-entropy text chunks (concepts), connected by high-entropy 'forking tokens' that represent decision points. The LLM reasoning process can be deconstructed into stable "concepts" (e.g., step 3) linked by high-entropy decision points. The probabilities at these junctures (e.g., "next" vs. "tens") represent transition weights on an underlying reasoning graph, a view supported by convergent findings on the role of high-entropy tokens. This figure is adapted from Ref. Cai et al. (2025).

Recent work, notably by (Minegishi et al., 2025), has developed detailed methods to empirically construct these task-specific reasoning graphs. Their process involves generating a large corpus of reasoning chains, segmenting them into individual steps, and then computing a vector representation for each step by averaging the hidden states of its tokens at a specific Transformer layer. Finally, a clustering algorithm (e.g., K-means) is used on the full set of step vectors, with the resulting cluster centroids being defined as the graph's nodes. By analyzing graphs constructed this way, they found that more advanced models exhibit distinct topological properties, such as larger diameters, which correlate with improved performance. However, the cluster centroids derived from K-means offer limited interpretability as stable reasoning concepts, making their correspondence to the models internal processes unclear. Moreover, the induced graph's structure is unstable, varying significantly with the choice of model, dataset, segmentation, and representation layer. Hence, this graph cannot track evolving representational dynamics during RLVR training process. How to construct a principled model of an LLM's internal reasoning process remains a formidable open problem.

While analyzing these static, task-specific graphs is useful, it leads to a deeper question: what is the underlying, universal knowledge structure that allows an LLM to generate so many different reasoning graphs across a multitude of tasks? It follows that these empirically observed reasoning graphs are best understood

as small projections of a much vaster, latent concept web. This web represents the entirety of the model's abstract knowledge, and the ultimate goal is to understand its structure.

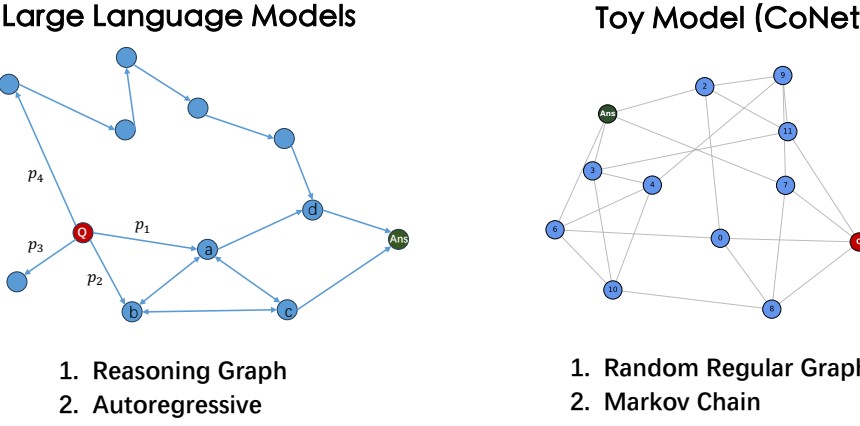

Figure 8: **From Intractable LLM Dynamics to a Minimal CoNet Model.** The reasoning process in an LLM (left) can be viewed as a traversal on a high-dimensional latent graph, where path probabilities are determined by the complex, autoregressive Transformer. Directly analyzing this graph's evolution is intractable. CoNet (right) provides a minimal abstraction by replacing the intractable latent graph with a fixed K-regular random graph and simplifying the generation policy to a learnable Markov chain. This creates a tractable "computational microscope" to precisely analyze how the transition policy evolves under reinforcement, isolating the core dynamics of reasoning path formation. This figure is adapted from Ref. Cai et al. (2025).

However, attempting to reverse-engineer this universal concept web from its sparse, task-specific projections presents formidable obstacles. The difficulties are multifold. First, there is an issue of scale: the true concept web is of astronomical size, and any extracted graph is a small sample. Second, the mapping itself is ambiguous: the mapping from a low-level token sequence to a high-level semantic concept is inherently many-to-one. Third, this already-ambiguous mapping is itself dynamic, as it can shift during RLVR training. These challenges collectively lead to the observable node stability problem in practice, making it nearly impossible with current methods to track the web's true topological evolution.

It is precisely this observational challenge that our paper bypasses with a two-pronged, indirect approach. First, we form a theoretical hypothesis about the consequences of a specific topological structure (i.e., sparsity). Second, we use a minimal model (CoNet) that captures the essential physics of the system to study these consequences in a controlled environment. Thus, while the direct construction of the concept web remains an open problem, our work proceeds from a central hypothesis about its fundamental structure. This hypothesis—that the web is profoundly sparse with an average degree near two—serves as the theoretical foundation for our paper. From it, we first demonstrate how this assumption of sparsity explains the known macroscopic dynamics of RLVR. We then use it to deduce previously overlooked microscopic mechanisms, namely frustration-induced forgetting and phase-transition-like learning. Finally, armed with these new insights, we explore how this deeper understanding can be leveraged to design a novel, improved RLVR algorithm.

## D    CoNet: A Minimal Model for RLVR dynamics

To bypass the intractable node stability problem, we leverage the Concept Network Model (CoNet). CoNet was originally introduced in (Cai et al., 2025) to study the dynamics of single-task RLVR, but its core design makes it perfectly suited for our investigation. By positing a fixed space of concept nodes (a static graph structure) and isolating all learning dynamics to the evolving policy that navigates it, CoNet allows us to sidestep the challenges of direct graph extraction and precisely measure how the reinforcement signal sculpts reasoning paths. The model represents the LLM's concept space as an abstract K-regular random graph, with the reasoning process simplified to a policy-guided Markovian random walk, as shown in Fig. 8. The transition policy, $\pi_\theta(j|i) \propto \theta_{ij}$, is governed by a set of learnable parameters $\theta_{ij}$ that represent the strength of the connection from node i to node j. This policy is updated via RLVR on successful paths from a question (Q) to an answer (A).

This design offers a clear cognitive analogy. A single-step hop on the graph represents "System 1" thinking—fast, intuitive associations. A multi-step traversal, guided by the learned policy, represents "System 2" reasoning—a deliberate, compositional chain of thought. It is crucial to note that CoNet is not a model of the Transformer's parameters themselves; rather, it models the emergent semantic network induced by inference—the structure that governs how questions are connected to answers through chains of concepts.

CoNet was originally introduced to model the phenomenon of Learning at Criticality (LaC), which was identified in single-task RLVR training (Cai et al., 2025). The central finding of LaC is that for a model trained on extremely sparse data (e.g., a single exemplar), generalization performance on unseen problems peaks precisely at a sharp, sigmoidal learning transition, before declining due to overfitting. The original work used CoNet to establish that this dynamic is a learning phase transition, providing a physical model for this peak in generalization.

The central methodological advance of this paper is to deploy this existing model in a richer, multi-task setting with numerous concurrent Q-A pairs. This extension is crucial: it transforms CoNet from a model of isolated skill acquisition into an ideal theoretical laboratory for studying skill integration. In this regime, the model is compelled by competing reinforcement signals to discover a shared, reusable structure. As Fig. 1 shows, this multi-task CoNet stunningly reproduces the key macroscopic signatures of the 1.5B LLM. This correspondence validates our use of CoNet as a minimal model for exploring the network-level mechanisms of frustration and web formation.

## E    CoNet Implementation Details

We configured CoNet with a specific set of structural and learning parameters. The underlying concept space was instantiated as a directed random regular graph with $N = 800$ nodes, where each node has a uniform out-degree of $k = 40$. In this graph, we established a multi-task learning environment consisting of 128 randomly generated Q-A node pairs. The transition parameters $\theta_{ij}$ for all edges were initialized randomly. The use of a multi-task setting with 128 concurrent Q-A pairs is the fundamental mechanism that induces structural competition. While a single task would only require finding one efficient path, the presence of numerous tasks provides competing reinforcement signals. When multiple Q-A paths need to traverse a common node, they may "disagree" on the optimal subsequent step, leading to conflicting policy updates. This microscopic conflict is the origin of the macroscopic "frustration" discussed in the main text, compelling the system to find a shared, globally coherent structure that can arbitrate these conflicts, rather than settling for 128 disconnected "skill islands".

The policy was optimized using a variant of the GRPO algorithm ((Shao et al., 2024; Liu et al., 2025; Yu et al., 2025)). For a given Q-A pair, $n_{\text{rollout}} = 128$ reasoning paths (indexed by $m$) were sampled, with

each path's exploration capped at a maximum length of 20 steps. Each path received a reward based on its success, and its advantage $A_m$ (relative to the average reward over all 128 rollouts) was used to guide the update of the policy parameters $\theta_{ij}$. The update rule is given by:

$$\Delta\theta_{ij} \propto \sum_m A_m \nabla_{\theta_{ij}} \log \pi_\theta(j|i)$$

This advantage-based update functions as a competitive reinforcement mechanism, strengthening above-average paths and suppressing underperforming ones. This "winner-take-more" dynamic acts as an algorithmic sculpting tool, systematically pruning the vast majority of the initial $k = 40$ edges from each node to carve out the sparse, efficient backbone of the concept web. The learning rate was set to $0.04$ to control the speed and stability of this training process.

Crucially, the two-stage learning dynamic is not an artifact of fine-tuned parameters but a robust emergent property of the model. Repeated experiments show that this behavior—a fast-learning phase followed by a slow integration plateau, and the corresponding V-shaped response length—consistently emerges under a broad condition: when the number of Q-A pairs, $N_{QA}$, is in the regime of $1 \ll N_{QA} \ll N$. This dynamic is qualitatively different from the single Q-A pair ($N_{QA} = 1$) case, which produces a simple sigmoidal learning transition (the LaC phenomenon). The condition is essential: if $N_{QA}$ is too small, there is no systemic competition to drive the formation of a unified web; if it is too large, the conflicting reinforcement signals prevent any coherent structure from forming. The robust emergence of these LLM-like signatures in this intermediate regime confirms that our CoNet setup successfully reproduces the essential dynamics of structural self-organization under multi-task reinforcement.

## F  STRUCTURAL EVOLUTION OF THE CONCEPT WEB IN CONET

This section provides direct, quantitative evidence from the CoNet model for the two-stage learning dynamic—a fast acquisition process followed by a slow integration phase—that underlies our theory. Fig. 9 serves as the central exhibit for this analysis, tracking three key structural properties over the course of training: the total number of disconnected solution clusters (the "skill islands"), the size of the single largest cluster, namely the concept web and the average degree of the concept web.

The first stage of training ($\sim$ 0-50 steps), corresponding to the fast-learning phase, is characterized by the rapid proliferation of "skill islands". The quantitative evidence for this is the steep, initial rise of the "Cluster Number" (orange curve) in Fig. 9. Each new cluster represents the discovery of a distinct, high-confidence reasoning path for a specific problem. During this phase, the system operates in a greedy, parallel fashion, finding many efficient, local solutions without regard for their interconnection.

The shift to the slow-learning phase occurs at the critical juncture marked by the peak of the orange curve around step 20. At this point, the system has exhausted the "low-hanging fruit" of simple, isolated solutions and has reached a state of maximal fragmentation. This peak signals the onset of the "maximally frustrated state" discussed in the main text. The subsequent, gradual decline of the "Cluster Number" curve is the direct signature of this frustrated state, representing the arduous and competitive process of integrating existing islands rather than discovering new ones. The "Max Cluster Size" (blue curve), which begins its steep, sustained ascent at precisely this moment, confirms this shift from local discovery to a global integration objective.

The slow-learning phase is thus defined by the coalescence of these islands into a unified concept web. The plot reveals the two complementary trends that define this stage: a steady decline in the total number of clusters (orange curve) and a simultaneous, relentless growth in the size of the largest cluster (blue curve). The causal link is direct: the largest component grows precisely by absorbing smaller, independent clusters, which necessarily reduces the total cluster count. Moreover, the average degree of the concept web remains

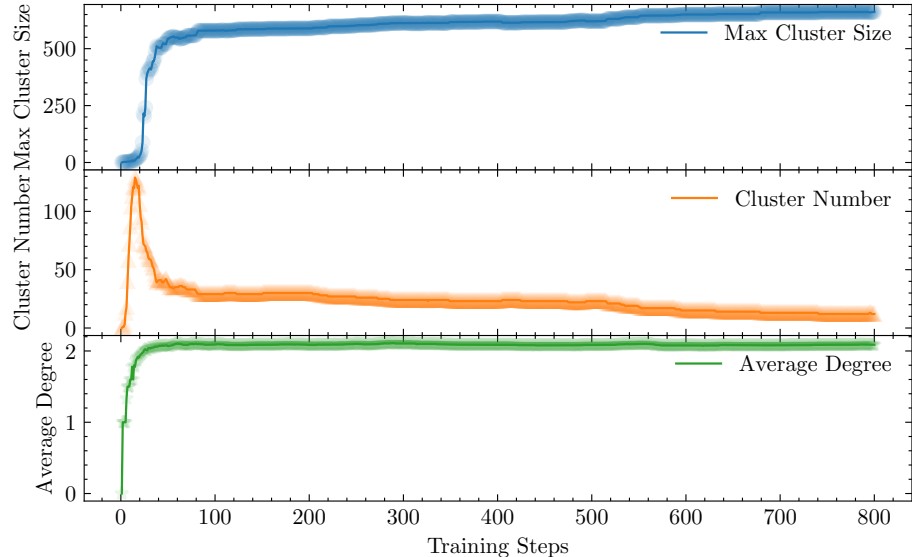

Figure 9: **The Formation of the Concept Web in CoNet.** This figure illustrates the evolution of network clusters and topology during training, showing a structural reorganization from isolated skills to a unified conceptual web. In the initial phase, the Cluster Number (orange curve) spikes, indicating the rapid discovery of numerous disconnected "skill islands". The peak marks a critical structural juncture. Following this peak, the Cluster Number steadily declines as these islands begin to integrate. This merging process is confirmed by the corresponding monotonic growth in the Max Cluster Size (blue curve), which represents the formation of a single giant component. Crucially, the Average Degree of the conept web (green curve) rapidly converges to and remains pinned at $\approx 2$ throughout the slow-learning phase. This persistent sparsity confirms that the global integration occurs within a strictly constrained, tree-like topology, validating our central hypothesis.

pinned at $\approx 2$ throughout the expansion, indicating that the concept web maintains a strictly sparse, tree-like topology. This structural evolution culminates in a state where a single, giant component—the functional concept web—dominates the network. This process provides a direct, mechanical explanation for the V-shaped response length signature: Stage 1 shortens paths by finding local solutions, while Stage 2 lengthens them by forcing the policy to traverse longer, connective paths between previously distant concepts.

## G  LEARNING BY PHASE TRANSITION IN THE SLOW-LEARNING STAGE

In the main text, we argue that new knowledge acquisition during the slow-learning stage is not gradual but occurs via discrete, sharp events. This appendix provides detailed, per-problem evidence for this learning by phase transition mechanism, drawing a direct analogy to critical phenomena in statistical physics.

Fig. 10 isolates a representative subset of problems mastered late in the training process. The first signature of a phase transition is visible in the left panel: each problem exhibits a sharp, sigmoidal-like jump in accuracy. In the language of statistical physics, accuracy acts as an order parameter, signaling the system's transition from a disordered (unsolved) state to an ordered (solved) state. An abrupt change in the order parameter is a primary indicator of a phase transition.

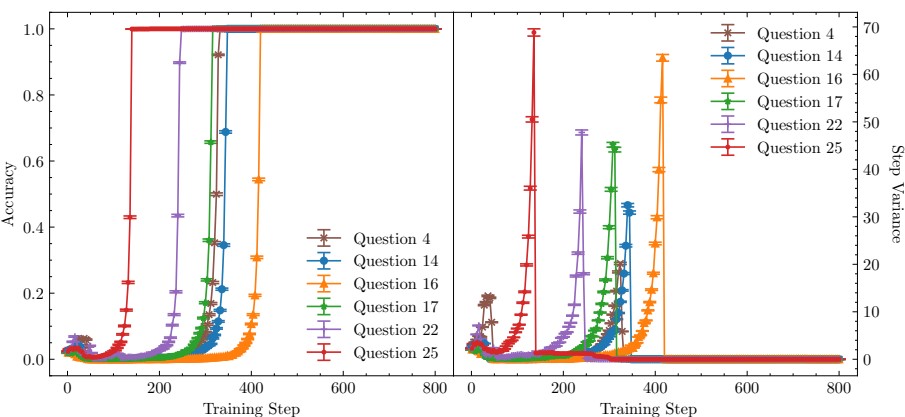

Figure 10: **Per-Problem Critical Dynamics During the Slow-Learning Stage.** These plots reveal the microscopic learning dynamics for a representative subset of problems that successfully learn during the slow-learning stage, providing evidence for localized phase transitions. **(Left)** Accuracy trajectories for these problems. All problems shown exhibit a sharp, sigmoidal-like increase in accuracy rate in the training process, characteristic of a critical learning event. **(Right)** Corresponding variance of the solution path length. A sharp, pronounced peak in variance directly coincides with the rapid accuracy gain for each problem. This dual signature is the hallmark of a continuous phase transition, suggesting that the slow-learning stage in CoNet is composed of discrete, sharp learning events rather than a uniform, gradual process.

The second, more telling signature is the pronounced, transient peak in the variance of the solution path length, shown in the right panel. This provides a deeper link to the physics of critical phenomena. In physical systems, the variance (or fluctuations) of an order parameter is related to the system's susceptibility—its response sensitivity to external perturbations. At the critical point of a continuous phase transition, this susceptibility is known to diverge, creating a characteristic sharp peak. A famous example of this is the lambda ($\lambda$) peak observed in the specific heat of liquid helium at the normal-to-superfluid transition. The variance peaks in our figure are the direct analogue of this phenomenon. This "dual signature"—an abrupt rise in the order parameter (accuracy) and a lambda-like peak in its fluctuations (variance)—is the classic hallmark of a critical learning transition. This same analysis was performed in the original work on LaC, where the variance peak in the single-task CoNet was identified as a key hallmark of a learning phase transition (Cai et al., 2025).

The intuitive reason for this variance peak is straightforward. When accuracy is very low, nearly all sampled paths fail, resulting in low variance. When accuracy is very high, the model has converged on a dominant, efficient reasoning path, again leading to low variance. It is only during the narrow transition window that a rich diversity of paths coexists—some still failing, while others explore newly viable routes to the solution. This temporary coexistence of competing strategies is what produces the characteristic peak in variance.

Taken together, these results provide strong microscopic validation for our claims. They show that the slow-learning stage in CoNet is not a process of uniform, gradual improvement. Instead, it is punctuated by a series of discrete, critical learning events, where individual skills are integrated into the concept web one by one via localized phase transitions at the frontier.

# H  THE SFT AND ANNEALED-RLVR ALGORITHM IN CONET

This appendix details the implementation and validation of the Annealed-RLVR algorithm in CoNet, our "computational microscope" for LLM reasoning dynamics. The algorithm is designed to intervene at the point of "maximum frustration" with SFT to resolve the competitive bottlenecks that arise during the formation of the sparse concept web.

## H.1  SFT IMPLEMENTATION IN CONET

CoNet's minimalistic policy model significantly simplifies the standard SFT process. This process involves first using the original model's policy to sample a successful reasoning path for a given problem. Then, for this selected path, we adjust its transition probabilities. For any transition on the path with a probability below a certain threshold, we raise it to that threshold value. To ensure that the probability distribution of each node remains valid, the probabilities of other outgoing transitions are proportionally reduced. The value of this target transition probability is a tunable parameter that determines the intervention's effect. For our annealing strategy, we set this value to 0.1. To induce catastrophic forgetting, we use a more aggressive value of 0.5.

## H.2  EXPEIMENTAL RESULTS

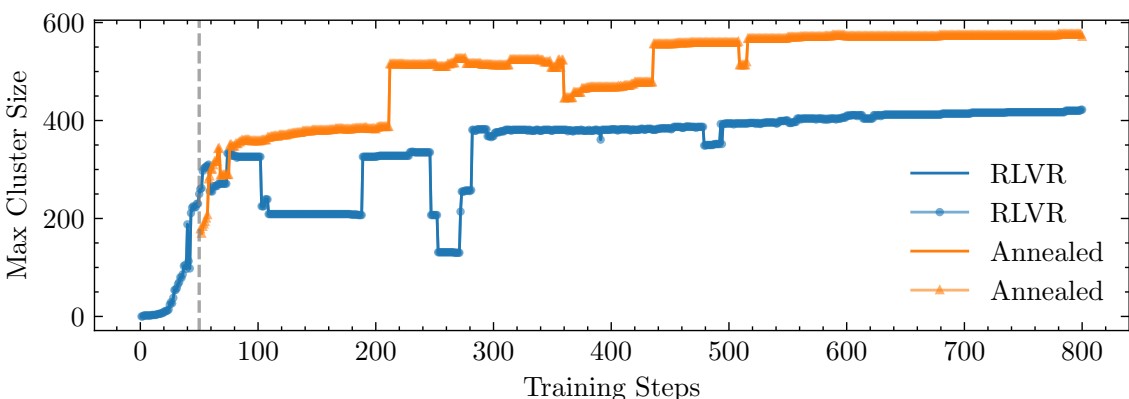

Figure 11: **Structural Impact of Annealed-RLVR on the Concept Web.** Evolution of concept web for standard RLVR (blue) versus Annealed-RLVR (orange). The SFT intervention at step 50 (dashed line) induces an immediate drop in the Annealed model's cluster size. Subsequently, the Annealed model recovers and surpasses the standard RLVR baseline, which exhibits slower growth, ultimately forming a larger final concept web.

The algorithm consists of a strategically timed SFT phase, followed by a resumption of standard RLVR training. The SFT intervention is applied at training step 50, the empirically determined state of maximum frustration, marking the crossover from the formation of disconnected "skill islands" to their subsequent integration. At this stage, we first identify all problems with an accuracy below 0.1, select a total of 50 low-accuracy problems, and sample a known successful reasoning path for each. This targeted adjustment serves as the "heating" step in our simulated annealing analogy; it gently increases the discoverability of latent correct paths to restore the system's exploratory capacity. Following this brief SFT, standard RLVR training is resumed, which functions as the "cooling" phase, allowing the now more exploratory policy to settle into a new, more globally optimal configuration.

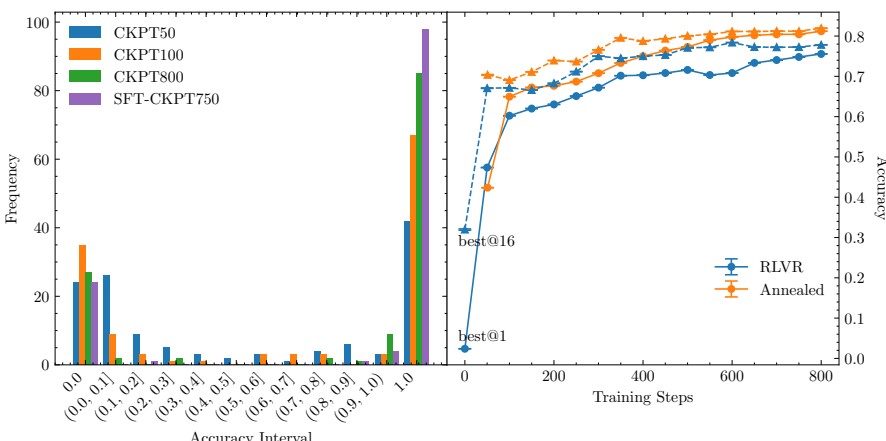

Figure 12: **Validation of Annealed-RLVR in CoNet. (Left)** The histogram of per-problem accuracy shows that the annealed model (SFT-CKPT750) significantly reduces the number of completely unsolved problems (Accuracy = 0.0) and increases the number of mastered problems (Accuracy = 1.0) compared to the standard model (CKPT800). **(Right)** The line plot of training dynamics shows that immediately after the SFT intervention at step 50, best@1 accuracy dips while best@16 accuracy sharply increases, indicating a successful trade of exploitation for exploration. The final annealed policy ("Annealed") ultimately achieves superior performance over the baseline ("RLVR").

Fig. 11 provides the direct structural evidence for this annealing process. As shown by the orange "Annealed" curve, the SFT intervention at step 50 locally disrupted network connections, resulting in a decline in the concept web size. However, this disruption is precisely what allows the system to overcome the integration bottleneck. While the standard RLVR model (blue curve) is shown to grow slowly, trapped in the maximally frustrated state, the Annealed model's cluster size rapidly recovers and grows to a larger final scale. This demonstrates that the SFT intervention, by resolving the competitive frustration, enables the subsequent RLVR "cooling" phase to successfully expand the network and integrate a greater number of "skill islands".

The success of this SFT-RLVR cycle is also confirmed by the results in Fig. 12. The line plot on the right shows the immediate effect of the intervention: best@1 accuracy (a proxy for exploitation) temporarily dips, while best@16 accuracy (a proxy for exploration) surges. This demonstrates a successful trade of greedy exploitation for enhanced exploration, and the long-term benefit is clear, as the final annealed policy significantly surpasses the baseline RLVR policy. The histogram on the left reveals the underlying mechanism for this improvement. The SFT-RLVR cycle successfully resolves the frustration bottleneck by reducing the population of completely unsolved problems (Accuracy = 0.0) and correspondingly increasing the number of fully mastered problems (Accuracy = 1.0) compared to the standard model.

In summary, the CoNet experiment provides strong evidence for our theory. By applying a targeted and conservative SFT intervention at the point of maximum frustration, the Annealed-RLVR algorithm effectively resolves the system's competitive bottlenecks, enabling it to discover a more robust and better reasoning policy.

# I LLM TRAINING DETAILS

## I.1 BASE MODEL AND GRPO SETUP

We used the **DeepSeek-R1-Distill-Qwen-1.5B** model (Guo et al., 2025), trained on the DeepScaleR-Preview-Dataset (Luo et al., 2025a) with their modified version of the veRL open-source engine (Sheng et al., 2024; ByteDance Seed Team and verl community). Our Group Relative Policy Optimization (GRPO) hyperparameters followed the first-stage settings, which specified a response length of 8192, as outlined in the DeepScaleR paper (Luo et al., 2025a).

## I.2 SFT DATASET FOR ANNEALED-RLVR

The SFT dataset for the **annealed-RLVR** experiment (labeled "Annealed" in Fig. 6) was constructed to focus on difficult problems. First, we trained the base model using GRPO for 100 steps to create a checkpoint (CKPT100), which is in the vicinity of the maximally frustrated state. We then used this checkpoint to generate 8 rollouts for each of the approximately 40,000 problems in the training set. For the roughly 10,000 problems that yielded no correct responses, we generated an additional 52 rollouts, bringing the total to 60 per problem. From this data, we created the SFT dataset by selecting all correct responses for problems where the initial model accuracy was below 10%, which yielded **3,933 trajectories**. Finally, we fine-tuned the model on these trajectories for two epochs with a learning rate of $3 \times 10^{-5}$. The resulting model served as the new starting point for continuing GRPO training up to step 1,700.

## I.3 SFT DATASET FOR CATASTROPHIC FORGETTING

The SFT dataset for the **catastrophic forgetting** experiment (labeled "SFT+RLVR" in Fig. 4(b)) was designed to specifically target problems forgotten by a more saturated model, using a data generation process that involved multiple checkpoints. The process began with problem filtering: we used a checkpoint from step 600 (CKPT600, deep within the slow-learning stage) to generate 8 rollouts for all training problems. We then identified the subset of problems with no correct responses and generated another 32 rollouts for them, again using CKPT600. Problems that still had no correct responses were selected as the final "forgotten" set. For the second stage of targeted data generation, each problem in this forgotten set was used to generate a total of 96 rollouts: 32 from the **base model**, 32 from CKPT100, and 32 from CKPT600. We then collected correct responses from problems that had at least 3 correct answers among these 96 rollouts, resulting in **798 trajectories**. This SFT dataset was used to fine-tune the model for two epochs with a learning rate of $5 \times 10^{-5}$ before resuming GRPO training.

## I.4 COMPUTATIONAL COST ANALYSIS

The overall computational complexity of **Annealed-RLVR** is dominated by the iterative GRPO training, making it nearly identical to the standard RLVR baseline. The only addition is a **one-time cost** for the SFT intervention, which is applied once at the point of maximal frustration (around step 100 in our case). This one-time cost consists of two phases:

1. **Evaluation Pass:** An evaluation pass over the $N$ problems in the training set (where $N \approx 40,000$) is required to identify the low-accuracy problems. As described in Section I.2, this involved generating 8 to 60 rollouts per problem. This cost can be formally described as $O(N \cdot k)$ for a fixed $k$ rollouts, or $O(N \cdot \log(k))$ if using an adaptive, binary-like rollout strategy to find problems below the accuracy threshold.

2. **SFT Phase:** The SFT itself is performed on the small subset of resulting correct trajectories. Letting $\eta$ be the accuracy threshold parameter (e.g., $\eta = 0.1$ for our 10% threshold), the SFT cost

is proportional to $O(\eta N)$. As detailed in Section I.2, this amounted to only **3,933 trajectories** (approximately $\eta N$) in our experiment.

This $O(\eta N)$ SFT cost is significantly smaller than the evaluation cost. Both of these marginal, one-time costs are much smaller than the total iterative cost of the full GRPO training run (e.g., 1,700 steps).

## J LLM EVALUATION DETAILS

We evaluated checkpoints from our RLVR ("GRPO") and annealed-RLVR ("Annealed") training runs on two datasets: 512 randomly selected training problems and the Minerva math dataset (Hendrycks et al., 2021; Lewkowycz et al., 2022). The evaluation parameters were kept consistent with the training configuration (temperature: 0.6, top-p: 1.0, response length: 8192, etc.). From the 512 rollout results, we calculated the best@k accuracy and the standard error. Fig. 6 illustrates the full performance curves, while Table 1 highlights the results from representative checkpoints.

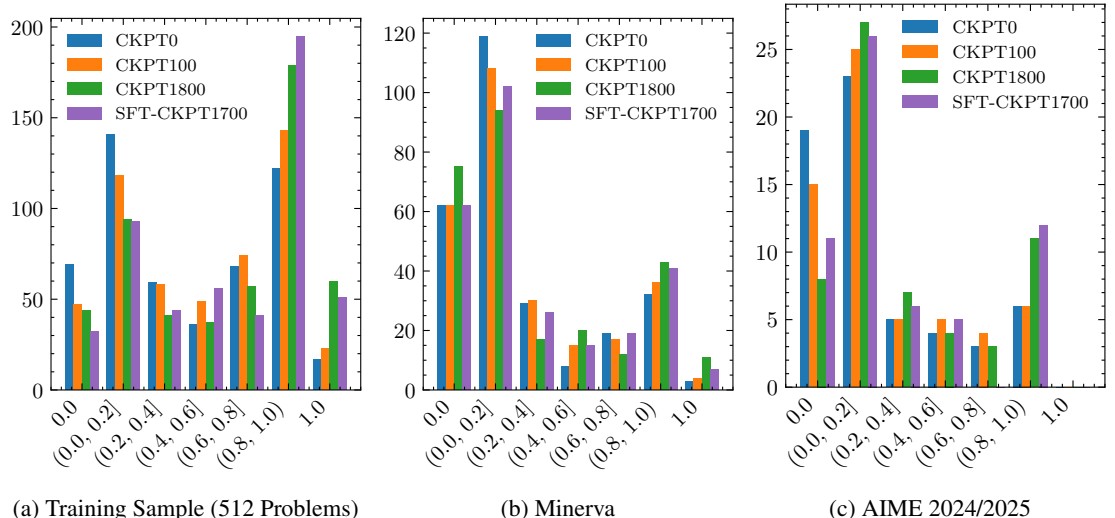

(a) Training Sample (512 Problems)  (b) Minerva  (c) AIME 2024/2025

Figure 13: **Mechanism of Annealed-RLVR: Accuracy Distributions.** This figure complements the best@k curves in the main text by revealing the underlying mechanism of performance improvement. It shows the per-problem accuracy histograms for the checkpoints of Annealed-RLVR ("Annealed") versus the standard RLVR ("RLVR") baseline, evaluated on (a) the in-distribution set (Random 512), (b) the OOD Minerva dataset, and (c) the OOD AIME 2024/2025 datasets. The histograms demonstrate that the Annealed-RLVR intervention systematically reduces the population of unsolved problems (low-accuracy bins) and increases the population of mastered problems (high-accuracy bins).

Table 1: Best@k Accuracy (%) with Standard Error at Key Training Checkpoints

| Metric | Method | Checkpoints (Steps) | | | | | | |
|---|---|---|---|---|---|---|---|---|
| | | 0 | 100 | 200 | 600 | 1000 | 1400 | 1800 |
| **Dataset: Randomly Selected 512 Training Problems** | | | | | | | | |
| best@1 | GRPO | 43.66(6) | 50.17(6) | 52.45(6) | 56.49(6) | 58.00(6) | 58.90(6) | 59.55(6) |
| | Annealed | — | 50.00(6) | 52.27(6) | 55.80(6) | 58.10(6) | 59.46(6) | **60.31(6)** |
| best@4 | GRPO | 61.32(8) | 68.46(8) | 70.38(8) | 72.99(7) | 73.46(7) | 73.71(7) | 74.20(7) |
| | Annealed | — | 69.49(8) | 71.51(8) | 73.75(7) | 75.10(7) | 75.91(7) | **76.77(7)** |
| best@16 | GRPO | 72.97(12) | 78.60(11) | 80.12(11) | 81.60(10) | 81.61(10) | 81.62(10) | 81.86(10) |
| | Annealed | — | 80.62(11) | 82.22(11) | 83.15(10) | 83.77(10) | 84.48(10) | **85.28(10)** |
| best@128 | GRPO | 82.16(24) | 86.98(23) | 87.81(20) | 87.47(19) | 88.23(21) | 87.77(19) | 87.99(22) |
| | Annealed | — | 88.00(19) | 89.35(20) | 89.37(19) | 89.98(18) | 90.23(18) | **91.22(18)** |
| **Dataset: Minerva** | | | | | | | | |
| best@1 | GRPO | 23.85(7) | 26.08(7) | 26.83(7) | 27.63(7) | 28.53(7) | 28.45(7) | 28.77(7) |
| | Annealed | — | 26.09(7) | 26.62(7) | 27.73(7) | 28.28(7) | 28.75(7) | **28.88(7)** |
| best@4 | GRPO | 37.84(12) | 40.77(12) | 41.33(12) | 41.12(12) | 41.46(11) | 41.04(11) | 41.32(11) |
| | Annealed | — | 40.63(12) | 40.90(12) | 41.66(12) | 42.04(11) | **42.53(11)** | 42.39(11) |
| best@16 | GRPO | 51.39(21) | **54.29(21)** | 54.20(21) | 53.05(20) | 52.38(19) | 52.12(20) | 51.95(19) |
| | Annealed | — | 53.80(21) | 53.71(21) | 53.74(20) | 53.86(21) | 54.24(21) | 53.50(20) |
| best@128 | GRPO | 66.6(5) | 68.8(4) | **69.8(5)** | 66.2(4) | 65.8(5) | 64.9(4) | 64.4(4) |
| | Annealed | — | 68.2(5) | 68.3(4) | 68.4(5) | 68.0(4) | 69.4(5) | 67.8(5) |
| **Dataset: AIME 2024 & 2025** | | | | | | | | |
| best@1 | GRPO | 20.11(0.10) | 23.08(0.11) | 24.04(0.12) | 26.98(0.12) | 27.77(0.11) | 28.11(0.11) | 28.10(0.11) |
| | Annealed | — | 23.27(0.11 | 23.86(0.12 | 26.77(0.12 | 27.67(0.11 | **28.26(0.11** | 27.93(0.11 |
| best@4 | GRPO | 33.31(0.17) | 38.44(0.19) | 40.33(0.19) | 42.74(0.18) | 43.36(0.19) | 43.51(0.18) | **43.88(0.18)** |
| | Annealed | — | 39.36(0.19) | 40.21(0.19) | 43.38(0.19) | 43.76(0.19) | 43.79(0.19) | 43.42(0.19) |
| best@16 | GRPO | 45.3(0.3) | 52.4(0.3) | 54.4(0.3) | 55.5(0.3) | 56.9(0.3) | 56.1(0.3) | 57.1(0.3) |
| | Annealed | — | 54.1(0.3) | 54.7(0.3) | 57.6(0.3) | 57.7(0.3) | **58.1(0.3)** | 57.8(0.3) |
| best@64 | GRPO | 55.9(0.5) | 62.4(0.5) | 64.2(0.5) | 64.5(0.6) | 65.9(0.6) | 64.8(0.6) | 66.2(0.6) |
| | Annealed | — | 63.8(0.5) | 64.2(0.5) | 67.0(0.5) | 66.7(0.5) | 67.5(0.5) | **67.7(0.5)** |

# K SUPPLEMENTARY EVIDENCE: SINGLE-TASK DYNAMICS

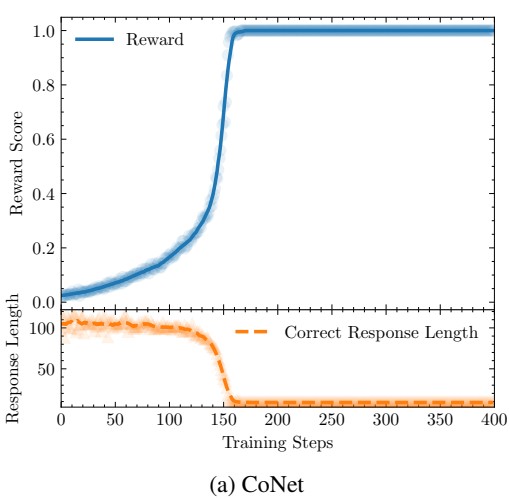 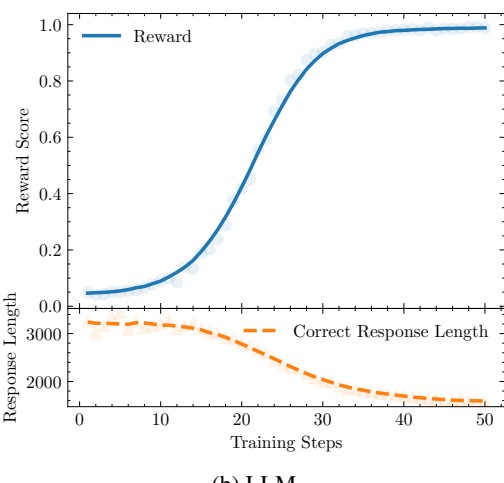

(a) CoNet                                   (b) LLM

Figure 14: **Single-Task Dynamics** We compare the training dynamics of CoNet (a) and the DeepScaleR-1.5B LLM (b) when trained on a single problem. The single-task experiment exhibits: (i) The reward curves (top panels) follow a sigmoidal, phase-transition-like trajectory; (ii) The correct response length (bottom panels) decreases monotonically, converging to the efficient paths. Lines in (a) and (b) are smoothed with a 5-step and 4-step moving average respectively.

We reproduced the single-problem training experiment studied by Cai et al. (2025) using the DeepScaleR protocol (Luo et al., 2025a). Specifically, we selected a single problem from the DeepScaleR-Preview-Dataset (Luo et al., 2025a) and repeated the single-task training experiment with their modified version of the veRL (Volcano Engine Reinforcement Learning). As shown in Fig. 14, the single-task dynamics exhibit a sigmoidal learning curve and a monotonically decreasing response length, standing in sharp contrast to the two-stage reward dynamic and V-shaped response length observed during multi-task training. CoNet faithfully reproduces these distinct behaviors when restricted to the same single-task condition.

This single-task training more effectively highlights the phase-transition-like learning mechanism described in the main text. By restricting the system to a single task, we isolate the fundamental phase-transition dynamics of skill acquisition: the model focuses solely on identifying an optimal trajectory among multiple feasible paths. Without the structural requirement to integrate conflicting skills into a shared, sparse backbone, the system avoids the "maximally frustrated state." Consequently, the response length decreases monotonically as the policy converges to a fixed optimal path, and the reward curve follows a smooth, sigmoidal trajectory characteristic of a continuous phase transition, free from competitive frustration. We also note that the reward curve is influenced by finite-size effects imposed by the maximum response length constraint, this phenomenon remains a key subject for further theoretical investigation.

This capacity to replicate the distinct phenomenologies of both conditions—from the phase transition of isolated learning to the competitive frustration inherent in multi-trajectory integration—provides strong evidence that CoNet serves as a minimal model for investigating the emergent mechanisms of RLVR.

