# OpenReview forum: "How LLMs Learn to Reason: A Complex Network Perspective"
_ICLR.cc/2026/Conference — Submitted to ICLR 2026_

### Official Review · Reviewer_sf2x · 2025-10-27

**Soundness:** 2
**Presentation:** 3
**Contribution:** 2
**Rating:** 2
**Confidence:** 3

**Summary:**

This paper proposes a theory of the learning trajectory of reasoning models, cast in terms of the evolution of a Markov process describing the transitions between semantic states during reasoning traces. The theory is supported by analysis of a proxy model, CoNet, that explicitly learns transitions in an artificial multitask path-learning setting. This model exhibits a phase transition from isolated paths to a large connected component that unifies tasks. This finding is used to explain two-stage piecewise-linear learning curves and V-shaped trajectories of response length as a function of training time. The paper then proposes an Annealed RLVR method that attempts to intervene at the point of the phase transition, selectively applying SFT to problems with low accuracy.

**Strengths:**

Potentially useful perspective on learning and reasoning dynamics

Novel variation on RLVR with good empirical results

**Weaknesses:**

Overall the evidence for the proposed theory is circumstantial and not so strong.

It’s tenuous to conclude that two models that produce two rather simple patterns as in figure 1 do so by the same underlying mechanisms.

The multitask CoNet and the artificial task it is trained on rely on many assumptions. I think more foundational work is required to establish multitask CoNet as a meaningful proxy for real LLMs before applying it to the questions addressed in the present paper. For example the learning dynamics in multitask CoNet hinge on resolving disagreements between different QA paths on the outgoing probabilities of nodes they intersect on (appx E paragraph 1). This hinges on the Markov property of the CoNet dynamics, which does not apply to LLMs. In an LLM earlier context disambiguates between different questions so that the model is not bound only by what it most recently output.

The paper assumes there is a single shift underlying the transition from decreasing to increasing responses and the transition from slow learning to fast learning, but these transitions don’t appear to coincide. Moreover the relative timing differs between the two models: In CoNet the bottom of the V is at about 40 training steps while the kink in the learning curve is at about 80. In the LLM these numbers look to be about 400 and 30.

Section 2 argues for a topological explanation of the phase of increasing response lengths, but I don’t see an explanation for the initial phase of decreasing response lengths, or for the two-phase learning curve.

Regarding the annealing proposal, extra training on the more difficult items seems like a pretty generic approach, so its success doesn’t necessarily support the theory.

**Questions:**

It would help to include early definitions of concept web, semantic state, and logical transitions.

What is the formal evidence that fig 2b really shows two stage learning? Learning curves are almost always concave like this.

Fig 3a would be easier to read if the distributions were normalized.

Why does a disconnected concept graph yield longer responses? The paper mentions traversing long intermediate paths, but what are those paths? More generally what is the relationship between traversing paths in a concept graph (which I assume happens during reasoning, not the final response) and the length of a response?

How do you formally determine when to intervene for annealing, i.e., when the maximally frustrated state occurs?

Is there any analysis of the network topology that supports the interpretation of how annealing helps?

---

> ### Author Response · Authors · 2025-11-21
>
> >Overall the evidence for the proposed theory is circumstantial and not so strong.
> >It’s tenuous to conclude that two models that produce two rather simple patterns as in figure 1 do so by the same underlying mechanisms.
>
> Our argument does not rely on matching two “simple patterns” in Fig. 1. Experimentally, we identify a four-part RLVR phenomenon package in a real 1.5B LLM, and we show that the same package appears in CoNet under RLVR. CoNet then provides structural observables (cluster structure, average degree, bridge edges) that explain this package via a sparse concept-web mechanism, and this mechanism yields a new intervention (Annealed-RLVR) that improves the 1.5B model. This combination of **multi-faceted correspondence + predictive mechanism** goes substantially beyond circumstantial pattern matching.
>
> **What we actually explain: a RLVR phenomenon package in the 1.5B LLM**
> The theory is aimed at explaining a concrete RLVR phenomenon package observed in the 1.5B LLM under RLVR and SFT:
> 1. Two-stage learning with V-shaped correct-response length
> A fast initial improvement followed by a slower integration phase, with the average length of correct responses first decreasing and then steadily increasing (Fig. 1(b)).
> 2. Catastrophic forgetting and fast recovery under SFT + RLVR
> A short SFT phase inserted into a converged RLVR model causes a sharp performance collapse, followed by rapid recovery when RLVR resumes (Fig. 4(b)).
> 3. Microscopic frustration and jump-like gains
> For individual problems, accuracy shows non-monotonic, frustration-like regressions and sharp, jump-like “phase-transition” learning dynamics (Fig. 5(b)).
> 4. Gains from Annealed-RLVR
> A brief SFT “heating” at the maximally frustrated stage followed by RLVR “cooling” improves both in-distribution and out-of-distribution performance, trading some immediate best@1 for better exploration and best@k (Fig. 6).
> These four phenomena are jointly present and consistent across tasks and runs; they are the target of our explanation, not just the two curves in Fig. 1.
>
> **Why the evidence goes beyond "circumstantial"**
> The key experimental fact is that the same RLVR phenomenon package (1–4) appears in CoNet under RLVR, even though CoNet is deliberately minimal (fixed sparse graph, synthetic multi-task QA, Markov dynamics):
> | Phenomenon from above (1–4) | CoNet | LLM |
> | --- | --- | --- |
> | Two-stage learning + V-shaped correct length | Fig. 1(a) | Fig. 2(a) |
> | Catastrophic forgetting + fast RLVR recovery | Fig. 4(a) | Fig. 4(b) |
> | Microscopic frustration + jump-like learning dynamics | Fig. 5(a)| Fig. 5(b) |
> | Annealed-RLVR gains| Fig. 12| Fig. 6 Fig. 13|
>
> CoNet then gives us structural observables that are inaccessible in the LLM:
> - we see many small "skill islands" merging into a giant connected component;
> - the average degree stabilizes near ≈ 2, yielding a sparse, near-tree backbone;
> - a few critical bridge edges control connectivity, and their deletion/repair under SFT/RLVR aligns with catastrophic forgetting and fast recovery.
>
> This sparse concept-web picture provides a single structural mechanism that:
> - explains why a two-stage learning curve and a V-shaped correct-response length arise generically when a sparse backbone forms;
> - explains why a short SFT phase can cause large collapses (breaking a few bridges) and why RLVR can repair them quickly;
> - explains why RLVR suppresses the exploration capability of LLMs;
> - identifies a maximally frustrated regime where an annealing-style intervention should be especially effective.
>
> Crucially, this mechanism then yields testable predictions for the 1.5B LLM (e.g., qualitative effect of SFT insertion, timing and form of the V-shape, benefit and tradeoffs of Annealed-RLVR), which we verify experimentally, including the observed improvements from Annealed-RLVR on both in-distribution and out-of-distribution benchmarks (Fig. 6). Thus, our claim is not "two models share one simple pattern in Fig. 1 ⇒ same mechanism." Rather, a single sparse concept web hypothesis-centered on a near-tree backbone with effective degree ≈ 2-accounts for a rich four-part phenomenon package in CoNet and correctly predicts how a real 1.5B model behaves under RLVR, SFT, and Annealed-RLVR. This, in our view, is substantially stronger than circumstantial evidence.

---

> ### Author Response · Authors · 2025-11-21
>
> >The multitask CoNet and the artificial task it is trained on rely on many assumptions. I think more foundational work is required to establish multitask CoNet as a meaningful proxy for real LLMs before applying it to the questions addressed in the present paper. For example the learning dynamics in multitask CoNet hinge on resolving disagreements between different QA paths on the outgoing probabilities of nodes they intersect on (appx E paragraph 1). This hinges on the Markov property of the CoNet dynamics, which does not apply to LLMs. In an LLM earlier context disambiguates between different questions so that the model is not bound only by what it most recently output.
>
> **Core point**
> Our central empirical finding is that a very minimal CoNet already reproduces the full RLVR phenomenon package we observe in a real 1.5B LLM (the four phenomena defined in Comment 1). This shows that, to explain these specific phenomena, it is sufficient to retain exactly these minimal elements; mechanisms present in LLMs but absent in CoNet (such as autoregressive context) are therefore not essential for the phenomena studied in this paper. The Markov/autoregressive issue is one concrete example of this more general conclusion.
>
> **CoNet’s minimal ingredients are sufficient**
> As discussed in first Comment, the 1.5B LLM under RLVR and SFT exhibits a RLVR phenomenon package. CoNet is constructed as a random regular graph featuring concept-level Markov dynamics and is trained via GRPO on multiple problems, intentionally removing all other complexities of LLMs. Empirically, this minimal CoNet still reproduces the same RLVR phenomenon package as the 1.5B LLM. The reproduction of both RLVR phenomenon package and single-task dynamics implies that the additional complexities of LLMs are unlikely to be *necessary* for explaining these phenomena. This is exactly why CoNet is meaningful as a proxy: it identifies a minimal set of structural ingredients that are experimentally sufficient to generate the observed behaviors and to motivate interventions (like Annealed-RLVR) that we then verify on the real 1.5B model.
>
> **Markov vs autoregressive as a concrete instance**
> The Markov/autoregressive issue is one specific instance of the above logic. We fully acknowledge that real LLMs display important beyond-Markov effects such as richer looping structures in the reasoning graph that CoNet does not attempt to model. Our claim is narrower: the experimentally observed CoNet–LLM correspondence indicates that such beyond-Markov effects are not the key drivers of the four RLVR phenomena we focus on. For this specific package of behaviors, the minimal CoNet ingredients are sufficient, which is why a deliberately simplified, Markov CoNet remains a useful proxy for understanding and intervening in RLVR training of large LLMs.
>
> >The paper assumes there is a single shift underlying the transition from decreasing to increasing responses and the transition from slow learning to fast learning, but these transitions don’t appear to coincide. Moreover the relative timing differs between the two models: In CoNet the bottom of the V is at about 40 training steps while the kink in the learning curve is at about 80. In the LLM these numbers look to be about 400 and 30.
>
> We clarify that the decrease→increase and slow→fast shifts likely represent a crossover between stages rather than a single, simultaneous transition. It is still an open question whether this process strictly qualifies as a phase transition, and this requires further theoretical investigation. Moreover, constrained by training parameters (e.g., batch size, rollout number), the training dynamics of the LLM exhibit fluctuations, which makes it difficult to define a precise "kink" point in the learning curve or a minimum point for the response length. For instance, the bottom of the response length for the LLM actually spans a range roughly between steps 200 and 400. Consequently, these features are better understood as broad crossover regimes rather than instantaneous events, rendering a precise temporal alignment between the two signals inherently fuzzy. To avoid ambiguity, we have replaced the term "transition" in the revised manuscript to avoid implying an instantaneous shift.

---

> ### Author Response · Authors · 2025-11-21
>
> >Section 2 argues for a topological explanation of the phase of increasing response lengths, but I don’t see an explanation for the initial phase of decreasing response lengths, or for the two-phase learning curve.
>
> We emphasize that Stage 1 is dominated by local optimization of simple tasks. Given the random initialization of transition probabilities, shorter paths inherently have a higher probability of being sampled. Consequently, the model preferentially optimizes towards these shorter response patterns. This dynamic drives the model to rapidly find short, efficient solutions for many individual problems, producing a proliferation of disconnected “skill islands.” This behavior is apparent in the network visualizations in Fig. 2(a),(b), which show many locally optimized paths early in training, and is quantitatively supported by the sharp rise in the number of solution clusters reported in Appendix F (Fig. 9).
>
> >Regarding the annealing proposal, extra training on the more difficult items seems like a pretty generic approach, so its success doesn’t necessarily support the theory.
>
> We argue our approach is distinct from a generic "train on hard items" strategy primarily due to its critical timing. A generic, late-stage SFT applied to a solidified policy is destructive: as our catastrophic forgetting experiments show (Fig. 4), this severs critical connections and triggers a performance collapse. In contrast, Annealed-RLVR applies a targeted "heating" step precisely at the point of maximal frustration, before the unified web has formed. The success of this specific timing---enhancing performance without the structural damage caused by late-stage interventions---differentiates it from generic approaches. Moreover, this experiment further supports our theory because this specific prediction (that intervention at this critical point improves outcomes) is validated consistently across both our minimal CoNet model and the large-scale LLM experiments.
>
>
> >It would help to include early definitions of concept web, semantic state, and logical transitions.
>
> We thank the reviewer for this suggestion. We clarify that "semantic states" and "logical transitions" are established concepts from the reasoning graph literature (cited in our Introduction), whereas the Concept Web is our specific theoretical contribution, defined centrally in Hypothesis 1 within the Introduction. To address the reviewer's point and ensure the literature terms are clearly understood in the context of our experiments, we have inserted a concise paragraph at the beginning of Section 2. This new text explicitly maps the abstract "semantic states" and "logical transitions" to the concrete components of our CoNet model (nodes and edges) . The Concept Web itself is then discussed in the immediately following text as the emergent largest connected component, ensuring a clear logical flow.
>
>
> >What is the formal evidence that fig 2b really shows two stage learning? Learning curves are almost always concave like this.
>
> The formal evidence for two-stage learning is the co-occurrence of two distinct macroscopic signatures within a narrow crossover regime, best identified by the minimum of the V-shaped response length curve (Fig. 1(b)), supplemented by the "knee" of the reward curve. The fact that the reward slows as the solution structure qualitatively changes (from shortening to lengthening) signals a crossover between distinct learning phases, moving from local optimization to global integration. Regarding the comment that learning curves are "almost always concave", we note this is not universal; single-problem training often exhibits non-concave, grokking-like transitions (Cai et al., 2025). This distinction is faithfully reproduced in our CoNet model (sigmoidal for single-task vs. concave for multi-task), which further reveals the microscopic phase transitions driving these dynamics.

---

> ### Author Response · Authors · 2025-11-21
>
> >Fig 3a would be easier to read if the distributions were normalized.
>
> We thank the reviewer for this suggestion. We have placed the normalized version of Fig. 3(a) as an inset, since the two versions of the plot convey complementary information. In the normalized view, we see more clearly that during the first stage (CKPT20 → CKPT50) the model primarily improves by increasing the fraction of short, locally optimized responses. By contrast, at CKPT800 the the probability density shifts toward longer responses, consistent with the expanding concept web enabling longer solution paths.
>
> However, we retain the unnormalized main plot because it encodes performance magnitude: since the rollout number is fixed, the histogram height directly reflects the total count of correct solutions. Crucially, the unnormalized view reveals that the absolute number of long responses remains nearly constant from CKPT20 to CKPT50. This clarifies that the high proportion of long responses at CKPT20 is simply due to the low total accuracy whereas the shift at CKPT800 represents the capability gain in traversing long paths induced by RLVR.
>
>
> >Why does a disconnected concept graph yield longer responses? The paper mentions traversing long intermediate paths, but what are those paths? More generally what is the relationship between traversing paths in a concept graph (which I assume happens during reasoning, not the final response) and the length of a response?
>
> We clarify that the increase in response length is driven by the emergent sparse, tree-like topology of the integrated concept web, not directly related to the connectivity. In CoNet, the intermediate path traversed from a Question node (Q) to an Answer node (A) is the model's generated reasoning chain (response).This corresponds to the Chain-of-Thought (CoT) in LLMs, and the path length on the graph is equivalent to the length of the response. In the sparse, tree-like concept web, as the scale of the web expands, the model gains the capability to solve complex Q-A pairs that require these longer responses.
>
> >How do you formally determine when to intervene for annealing, i.e., when the maximally frustrated state occurs?
>
> We thank the reviewer for this important question. Based on our theory, the "maximally frustrated state" coincides with the crossover from the fast-learning phase to the slow-learning plateau. Therefore, we propose a general heuristic for timing the intervention: it should be applied when one monitors the key macroscopic signature of this transition---the bottom of the V-shaped response length curve (Fig. 1). This point marks the moment where the average solution length stops its initial decrease (driven by local optimization) and begins to steadily increase (driven by global integration). We find that this signature is operationally straightforward to monitor. Furthermore, precise timing is not strictly required; the intervention remains effective as long as it is applied within the general vicinity of this crossover.
>
> >Is there any analysis of the network topology that supports the interpretation of how annealing helps?
>
> Thank you for the question; we add a direct topological analysis in appendix H through tracking the evolution of concept web (Fig. 11). The maximum cluster size in Annealed experiment (orange curve) shows a sharp drop after the SFT intervention at step 50, followed by a rapid increase during subsequent RLVR training. Notably, the cluster eventually grows to a larger final scale than that of the standard RLVR model (blue curve). This phenomenon reflects that the SFT intervention at the maximally frustrated state has a limited topological impact on the concept web structure, while effectively restoring connections previously suppressed by frustration. This allows the model to partially escape the competitive bottleneck that traps the standard RLVR training.

---

> ### Author Response · Authors · 2025-12-03
>
> >What is the formal evidence that fig 2b really shows two stage learning? Learning curves are almost always concave like this.
>
> To provide concrete evidence supporting our point that learning curves are not universally concave, we reproduced a single-problem training experiment using the DeepScaleR protocol and dataset (refer to Appendix K). As detailed in the appendix, the single-task dynamics yield a clear sigmoidal learning curve rather than a concave one, accompanied by a monotonically decreasing response length. This stands in contrast to the concave reward curve and V-shaped response length observed during multi-task training. Importantly, the capacity to replicate the distinct phenomenologies of both conditions---specifically the phase transition observed in single-problem learning and the phenomenon package of multi-problem training---provides strong evidence that CoNet serves as a minimal model for investigating the emergent mechanisms of RLVR.

---

### Official Review · Reviewer_w85W · 2025-10-31

**Soundness:** 4
**Presentation:** 3
**Contribution:** 4
**Rating:** 10
**Confidence:** 4

**Summary:**

The paper provides a complex networks perspective to elucidate the behavior of reasoning models during RLVR training. As a proxy, the paper uses Co-NET which was shown to replicate the training dynamics of large scale LLM model.  I think the paper is very good, a little hard to read, but, the novelty contribution and explanations are excellent.

**Strengths:**

The paper in my opinion is phenomenal in its clarity and its contribution.
The theoretical insights at the beginning of the paper are clear, concise and strongly motivate the algorithm at the end.

**Weaknesses:**

- Not enough experiments on the algorithm.
- Compuational complexity of the algorithm is not studied.
- I wold like some mathematical modeling of these phenomenons, carefully modelled with precise theorems and lemma. I understand this could be a future work and will look forward to that paper.
- At many places in the paper, the English is very strong, the use of obscure words is common, I am okay with it, but, such strong english might reduce accessibility to the paper. For example, this sentence

*Our sparse web theory reveals that the smooth, aggregate reward curve of RLVR training masks a dramatic
microscopic behavior. Individual problem trajectories in Fig. 5 reveal a fundamental tension born from the
network’s sparse topology: a subtractive process of frustration-induced forgetting, peaking at the onset of
slow learning, and an opposing, additive process of phase-transition-like learning that drives knowledge
integration throughout the slow-learning stage.*

I mean, it took me a few seconds to grasp this. For better readability, i recommend making the prose a little lighter. This is novel level prose.

**Questions:**

- Does the expansive concept web emerge as a property or something changes in the optimization procedure to attain this and how is this property of generating a complex web related to overfitting which would potentialy lead to forgetting.

- When RLVR training resolders the severed connection by the SFT, does it not lead to a push-pull type mechanism, which might deteriorate the learning effectiveness in the long run? Doe you think there is a relationship between this dynamic and the frustration-phase transition dynamic you talk about later in the paper.

- It is curious to see that, the more complex the connectivity gets, the better the model responds to a variety of prompts? Intuitively, I would think that a denser graph is probably more amenable to better generaliztaion, is there any link between the two, that you found in your study?

- the annealing required for RLVR, how does it relate to the discount factors probably present in the GRPO being used in the underlying training and how is the annealing designed?

---

> ### Author Response · Authors · 2025-11-21
>
> > Not enough experiments on the algorithm.
>
> To broaden our experimental validation, we have added a new evaluation set, AIME 2024/2025, and included these results in the revised manuscript (Fig. 6, Fig. 13 and Table 1).
>
> >Compuational complexity of the algorithm is not studied.
>
> We clarified the computational cost in appendices. The overall complexity of Annealed-RLVR is dominated by the RLVR (GRPO) training, making it nearly identical to the standard baseline. The only addition is a one-time cost for the SFT intervention, which consists of (1) an evaluation pass over the N problems in the training set to identify low-accuracy problems and (2) the SFT itself on the resulting correct trajectories . The SFT cost (part 2) is proportional to O(N/m), where m is the inverse of the accuracy threshold (e.g., m=10 for our 10% threshold), as it trains on only a small fraction of the data (e.g., our 3,933 trajectories from N≈40,000 problems, as detailed in Appendix I). The evaluation cost (part 1) can be implemented efficiently; for example, using an adaptive, binary-like rollout strategy to find problems below the accuracy threshold with k rollouts, this cost is O(N*log(k)). This O(N/m) SFT cost is significantly smaller than the O(N⋅log(k)) evaluation cost, and both marginal one-time costs are dwarfed by the total, iterative cost of the full RLVR training run. We added this detailed cost analysis to the appendix (I.4) in our revised manuscript.
>
> >I wold like some mathematical modeling of these phenomenons, carefully modelled with precise theorems and lemma. I understand this could be a future work and will look forward to that paper.
>
> We fully agree that establishing a rigorous mathematical foundation with precise theorems and lemmas is the ultimate goal for understanding these phenomena. While CoNet is intended to be a minimal model, rigorous mathematical derivation remains challenging. Specifically, we have not yet found an analytically solvable model that can be effectively mapped to CoNet to aid in theoretically understanding these problems.
>
>
> >At many places in the paper, the English is very strong, the use of obscure words is common, I am okay with it, but, such strong english might reduce accessibility to the paper. For example, this sentence
>
> We thank the Referee for this valuable suggestion. Following the advice, we have carefully revised the manuscript to simplify the language, reduce the use of obscure vocabulary, and improve overall readability. We believe the presentation is now more accessible and less strong while preserving the precision of the scientific content.
>
> >Does the expansive concept web emerge as a property or something changes in the optimization procedure to attain this and how is this property of generating a complex web related to overfitting which would potentialy lead to forgetting.
>
> The emergence of the sparse, expansive concept web is a direct consequence of the intrinsic optimization dynamics of RLVR. This structure self-organizes during the slow-learning phase (Stage 2), where numerous isolated “skill islands” aggregate and connect, forming a sparse, tree-like concept web. Navigating this web to derive solutions for new problems expands its scale.
>
> At the microscopic level, the frustration-induced forgetting mechanism and the phase-transition-like learning dynamics described in Section 4 clarify how this emergent structure is tied to overfitting-like forgetting. When multiple paths compete, the process reinforces the probability of some edges while weakening the others, leading to frustration-induced forgetting.
>
> In parallel, phase-transition learning dynamic implies that once a particular QA pair crosses the transition point, its reasoning trajetories collapses onto optimal trajectories---policy collapse caused by overfitting. Previous work (Cai et al., 2025) explicitly demonstrated the overfitting phenomenon (loss of generalization) induced by phase-transition learning in single Q-A training.
>
> Taken together, these observations show that the complex, sparsely connected concept web is not engineered but rather emerges from the RLVR procedure itself, and its evolution is naturally coupled to the overfitting-like forgetting patterns characteristic of prolonged RLVR training.

---

> ### Author Response · Authors · 2025-11-21
>
> >When RLVR training resolders the severed connection by the SFT, does it not lead to a push-pull type mechanism, which might deteriorate the learning effectiveness in the long run? Doe you think there is a relationship between this dynamic and the frustration-phase transition dynamic you talk about later in the paper.
>
> We consider two cases regarding the 'SFT' mentioned in the question. If referring to the late-stage SFT, we agree with the reviewer's intuition: as shown in our catastrophic forgetting experiments (Fig. 4), such an intervention severs critical connections and indeed deteriorates learning effectiveness in the long run.
>
> In contrast, the early-stage Annealed-RLVR intervention is intentionally small-scale, targeted at low-accuracy problems, and timed precisely before the web solidifies---at the crossover regime of maximal frustration. Because the unified concept web has not yet formed, the SFT 'damage' is minimal. Our evaluation results over the full 1800 steps strongly support this, showing that Annealed-RLVR achieves superior long-term performance on both in-distribution (Fig. 6(a)) and out-of-distribution (Minerva and AIME24/25, Fig. 6(b) and (c)) datasets compared to standard RLVR. We thus do not expect the early stage SFT in Annealed-RLVR to deteriorate long-term learning.
>
> To the second point, there is a direct and positive relationship between the SFT dynamics and the frustration-phase transition dynamics: our SFT step is designed to alleviate the frustration. In systems with disorder, frustration arises from competing constraints that prevent the system from settling into a globally optimized state; this is analogous to our "maximally frustrated state" where skill islands compete for limited connections available in the sparse concept web. Our SFT "heating" step acts like the "heating" in simulated annealing : it provides a corrective jolt, allowing the system to escape this competitive bottleneck (a sub-optimal local minimum). By boosting the policy for difficult, low-accuracy paths, it prevents the standard RLVR "cooling" phase from prematurely "killing" these potentially valuable solutions, thus resolving the bottleneck and guiding the system toward a more robust, globally optimal configuration.
>
> >It is curious to see that, the more complex the connectivity gets, the better the model responds to a variety of prompts? Intuitively, I would think that a denser graph is probably more amenable to better generaliztaion, is there any link between the two, that you found in your study?
>
> We share your view that models with more complex connectivity are better equipped to handle a range of prompts. We also agree that greater connectivity enhances generalization, yet this typically compromises accuracy. How to optimally balance this trade-off remains an open question for future research. Turning to the current RLVR paradigm, our experiments reveal the concept webs produced by RLVR---particularly under the GRPO optimization---tend to be sparse rather than dense. Given this topological property, generalization arises from the expansion of this sparse, tree-like concept web: as its scale grows, the system can incorporate a larger set of concepts and traverse more diverse solution pathways.
>
> >the annealing required for RLVR, how does it relate to the discount factors probably present in the GRPO being used in the underlying training and how is the annealing designed?
>
> To clarify the reviewer's question, our GRPO setup follows the commonly used configuration (say, DeepScaleR and DeepSeek R1), where the discount factor is set to 1 (i.e., no discount). The term "annealing" in our work refers to the design of the Annealed-RLVR algorithm itself: the process of applying a single, targeted SFT "heating" shock at the point of maximal frustration, followed by resuming standard RLVR as a "cooling" phase. The potential interplay between different discount factors and this annealing dynamic is an interesting question for future investigation.

---

> > ### Comment · Reviewer_w85W · 2025-11-26
> > **thank you**
> >
> > Thank you for your responses, they make sense. This is a beautiful paper, I enjoyed reading it, I hope it gets accepted here.

---

### Official Review · Reviewer_jYxa · 2025-11-03

**Soundness:** 2
**Presentation:** 2
**Contribution:** 2
**Rating:** 4
**Confidence:** 3

**Summary:**

The paper proposes to explain RLVR’s distinct behaviors (e.g., two-stage learning, V-shaped response lengths, and catastrophic forgetting) through a unified theory of a sparse concept web. This topology unifies learning and forgetting: early “skill islands” compete under sparsity, causing frustration-induced forgetting, followed by phase-transition-like growth as new skills are integrated. Based on this insight, the authors propose Annealed-RLVR, which introduces a brief SFT phase to overcome bottlenecks before resuming RLVR. Experiments on a 1.5B model show improved reasoning and generalization over standard RLVR.

**Strengths:**

- The paper offers a unique and conceptually rich framework that connects RLVR training dynamics with complex network theory. By modeling reasoning as the self-organization of a sparse concept web, it provides an interesting explanation for several long-observed yet poorly understood phenomena in RLVR.

- The proposed sparse-web hypothesis is also accompanied by empirical illustrations such as the compelling visualizations that align well with the theoretical predictions and empirical RLVR training dynamics.

**Weaknesses:**

- The paper’s central motivation is not clearly articulated. While it introduces a novel complex-network perspective on RLVR training, it does not sufficiently explain why existing explanations are inadequate or how/where their framing offers better practical or conceptual advantages.

- The most direct and actionable contribution is perhaps the proposed "Annealed-RLVR." Though intriguing, the experiments are only conducted on a single 1.5B-parameter model across two datasets (one of which is questionable, as discussed in the questions section). The limited experiments restricted to one model lack broader validation to verify the empirical implications and generalization of the proposed sparse-web hypothesis. In addition, many essential experimental details (e.g., the LM adopted, the training/evaluation datasets used, baselines compared with) are omitted in the main content.

- The paper heavily relies on analogies and terminology from physics and complex network theory without providing sufficient background knowledge or clear empirical mappings between the proposed sparse concept web and measurable LLM internals. While theoretically appealing, this abstraction makes it difficult for readers without prior background in complex networks to follow, making the paper dense and quite challenging to read.

**Questions:**

- For the in-distribution experiments, the evaluation dataset is described as “randomly selected 512 training problems.” Are these examples part of the same training set used during Annealed-RLVR training? If so, how do you distinguish between training and evaluation performance in these results and potential contamination issues?

---

> ### Author Response · Authors · 2025-11-21
>
> > The paper’s central motivation is not clearly articulated. While it introduces a novel complex-network perspective on RLVR training, it does not sufficiently explain why existing explanations are inadequate or how/where their framing offers better practical or conceptual advantages.
>
> We thank the reviewer for this constructive comment. In the revised manuscript, we have substantially clarified the paper’s central motivation by rewriting the opening paragraph of the Introduction. The new paragraph explicitly details four distinct phenomena observed in RLVR training, highlighting how they are currently examined through separate theoretical lenses: (i) the learning plateau and (iv) policy collapse, predominantly studied via entropy dynamics; (ii) the V-shaped response length trajectory, often attributed to redundancy pruning or length-shifting heuristics; and (iii) catastrophic forgetting, typically analyzed as an alignment or objective-mismatch issue .
>
> We then argue that existing explanations, while valuable, remain fragmented because they address these phenomena in isolation. We provide two examples of this limitation:
>
> 1. Research on policy collapse primarily focuses on entropy dynamics to preserve the model's exploration ability. Concurrently, recent high-performing protocols (e.g., DeepScaleR, Skywork) emphasize the V-shaped length trajectory---where solutions first shorten and then lengthen---as a signature of improved reasoning performance. However, there is currently no theoretical framework connecting them. We lack an understanding of how the evolution of the response (the V-shape) interacts with the internal state of the policy (entropy and exploration). Specifically, existing theories do not explain why the specific dynamical path required for reasoning success (the V-shape) might inherently drive the system toward the specific internal states associated with collapse or frustration.
> 2. Similarly, attributing catastrophic forgetting solely to an objective mismatch predicts that forgetting will occur, but does not fully explain the extra fragility of the policy specifically during the slow-learning plateau (phenomenon i)---i.e., why the model becomes more susceptible to collapse at that specific stage.
> Our revision clarifies that a core contribution of this paper is to propose a unifying framework that ties these co-occurring behaviors to a single underlying process of structural self-organization. This framing aims to complement existing case-by-case analyses with a cohesive theory, helping to advance toward a more predictable engineering of emergent reasoning.
>
> > The most direct and actionable contribution is perhaps the proposed "Annealed-RLVR." Though intriguing, the experiments are only conducted on a single 1.5B-parameter model across two datasets (one of which is questionable, as discussed in the questions section). The limited experiments restricted to one model lack broader validation to verify the empirical implications and generalization of the proposed sparse-web hypothesis.
>
> To broaden our evaluation and further test the generalization of our findings, we have taken two steps:
> To address the "two datasets" concern, we have added a third evaluation set: the out-of-distribution (OOD) AIME 2024/2025 datasets. These problems were not part of the training data. Our results, which were included in the revised manuscript (Fig. 6, Fig. 13 and Table 1), show that Annealed-RLVR outperforms the baseline on this new OOD set as well, further strengthening our empirical claims.
>
> >  In addition, many essential experimental details (e.g., the LM adopted, the training/evaluation datasets used, baselines compared with) are omitted in the main content.
>
> We moved the essential experimental details from the appendices (previously I and J) to the main text to improve clarity. To explicitly address the reviewer's points, we state that: 1. The LLM adopted is the DeepSeek-R1-Distill-Qwen-1.5B model. 2. We used the DeepScaleR-Preview-Dataset for training. We evaluated performance on an in-distribution set (512 randomly selected training problems) and out-of-distribution sets (the Minerva dataset, plus AIME 2024/2025 results which were added to the manuscript in Fig. 6, Fig. 13 and Table 1). 3. The baseline compared with is the standard RLVR (GRPO) baseline (following the DeepScaleR configuration), which we evaluate against our proposed Annealed-RLVR.

---

> ### Author Response · Authors · 2025-11-21
>
> > The paper heavily relies on analogies and terminology from physics and complex network theory without providing sufficient background knowledge or clear empirical mappings between the proposed sparse concept web and measurable LLM internals. While theoretically appealing, this abstraction makes it difficult for readers without prior background in complex networks to follow, making the paper dense and quite challenging to read.
>
> We thank the reviewer for this feedback on readability. To make our physics-based abstractions more accessible, we have made three key revisions:
> 1. Clarifying Physics Terminology: In the Introduction, we added concise, intuitive definitions for our core physical metaphors. We now explicitly define "frustration" as an intense competition where the integration of some new skills forces the suppression of others, and "phase-transition-like learning" as a dynamic manifesting as capability jumps for individual questions. We hope this help readers to follow the narrative without prior physics background.
> 2. Mapping Abstract Concepts: In Section 2, we added a paragraph that explicitly maps the general literature terms ("semantic states," "logical transitions") to the concrete components of our CoNet model (nodes and edges).
> 3. Motivating the Abstraction: We revised the Introduction to clarify that we use this RG-inspired abstraction specifically to bypass the "formidable open problem" of empirically mapping LLM internals. This framing explains why a high-level theoretical model is necessary to provide a unified explanation for the observed macroscopic behaviors.
>
> We hope these revisions make the paper's logic and terminology much easier to follow.
>
> > For the in-distribution experiments, the evaluation dataset is described as “randomly selected 512 training problems.” Are these examples part of the same training set used during Annealed-RLVR training? If so, how do you distinguish between training and evaluation performance in these results and potential contamination issues?
>
> To clarify this point, the 512 problems are indeed randomly sampled from the full ~40,000-problem training set. The purpose of this dataset is to serve as a representative sample for monitoring the model's training dynamics with high precision (see Fig. 6 and Fig. 13). We assess true generalization performance using out-of-distribution (OOD) data. Our primary OOD sets are the Minerva dataset and the newly added AIME 2024/2025 problems. These datasets are not only excluded from the training set (which stops at AIME 2023) but also represent a higher difficulty level, as evidenced by the lower baseline accuracy compared to the in-distribution set. We have made this distinction between monitoring training performance (512-problem sample) and evaluating OOD generalization on these harder benchmarks clearer in the revision.

---

### Meta-Review · Area_Chair_sEqT · 2025-12-24

**Summary:**

This paper proposes a "Concept Network Model" (CoNet) to unify disjoint phenomena observed in RLVR training, such as the V-shaped response length, two-stage learning, and catastrophic forgetting, under a single theoretical framework based on sparse network topology ($\langle k \rangle \approx 2$). By identifying a "maximally frustrated state"  during the training dynamics, the authors derive "Annealed-RLVR," a theoretically motivated intervention that injects a Supervised Fine-Tuning (SFT) step at this critical bottleneck to improve exploration. While initial reception was polarized regarding the validity of using a minimal proxy model for complex LLMs, the rebuttal strengthened the submission by including out-of-distribution benchmarks like AIME 2024/2025.

The AC recognizes the paper’s novel theoretical clarity and contributions. However, the AC believes that the analogy between CoNet and LLMs requires further study to confirm a mechanistic parallel, as the current link remains largely phenomenological (Reviewer sf2x). Furthermore, the proposed annealing algorithm does not appear to generate substantial gains on key benchmarks like Minerva and AIME, and a more thorough sweep of annealing timing is needed to firmly establish the intervention's efficacy. Finally, the empirical validation would be significantly strengthened by evaluating a broader range of model classes beyond a single 1.5B-parameter model (Reviewer jYxa). Weighing these concerns, the AC’s recommendation is Reject, but we encourage the authors to incorporate this feedback for a future submission.

**Reviewer Concerns:**

Addressed Concerns:
- Limited Experimental Scope (jYxa, w85W): The authors strengthened their empirical validation by adding a third, out-of-distribution evaluation set (AIME 2024/2025). This directly addressed the concern that results were restricted to limited datasets.
- Generic Nature of Algorithm (sf2x): The reviewer questioned if the annealing was just a generic "train on hard items" strategy. The authors effectively clarified that timing is critical: late-stage SFT causes catastrophic forgetting (as shown in experiments), while early-stage "annealing" at the frustrated state succeeds. This specific temporal prediction validates the theory beyond generic heuristics.
- Computational Complexity (w85W): The authors added a detailed analysis showing the SFT cost is marginal ($O(N/m)$) compared to the dominant RLVR training cost.
- Motivation and Readability (jYxa): The authors rewrote the introduction to explicitly frame the "four-part phenomenon package" and defined physics-heavy terms like "frustration" to improve accessibility for a general ML audience.

Outstanding Concerns:
- Limited Experimental Scope (jYxa, w85W): The request to experiment beyond a single 1.5B-parameter model is not incorporated during the rebuttal phase.
- Proxy Model Validity (sf2x): While the authors argue that the minimal CoNet model is sufficient because it reproduces the complex behaviors of the LLM , the fundamental objection that LLMs are autoregressive while CoNet is Markovian remains a theoretical gap. Skeptical reviewers may view the correspondence as phenomenological rather than strictly causal.
- Mathematical Rigor (w85W): The request for precise theorems and lemmas was acknowledged by the authors as a goal for future work rather than being resolved in this revision.

**Reviewer Scores:**

Reviewer jYxa (4 to 5): This reviewer likely would have moved to a borderline accept. Their primary critique regarding limited experiments was directly addressed by the inclusion of the AIME benchmark, and the readability improvements addressed their concerns about the paper's density.

Reviewer w85W (10 to 10): This reviewer was already enthusiastic. The addition of the complexity analysis and extra experiments would reaffirm their strong support.

Reviewer sf2x (2 to 3): This reviewer might have softened their stance but likely remained skeptical. The authors provided a strong defense regarding the timing of the intervention as proof of the theory (distinguishing it from generic SFT), which counters the "generic approach" critique. However, the fundamental disagreement about using a Markovian graph as a proxy for Transformers is difficult to fully overcome in a rebuttal.

---

### Decision · Program_Chairs · 2026-01-26

Reject